# IMPLICIT BIAS OF SGD IN $L_2$-REGULARIZED LINEAR DNNS: ONE-WAY JUMPS FROM HIGH TO LOW RANK

**Zihan Wang & Arthur Jacot**
Courant Institute of Mathematical Sciences
New York University
New York, NY 10012, USA
`{zw3508,arthur.jacot}@nyu.edu`

## ABSTRACT

The $L_2$-regularized loss of Deep Linear Networks (DLNs) with more than one hidden layers has multiple local minima, corresponding to matrices with different ranks. In tasks such as matrix completion, the goal is to converge to the local minimum with the smallest rank that still fits the training data. While rank-underestimating minima can be avoided since they do not fit the data, GD might get stuck at rank-overestimating minima. We show that with SGD, there is always a probability to jump from a higher rank critical point to a lower rank one, but the probability of jumping back is zero. More precisely, we define a sequence of sets $B_1 \subset B_2 \subset \cdots \subset B_R$ so that $B_r$ contains all critical points of rank $r$ or less (and not more) that are absorbing for small enough ridge parameters $\lambda$ and learning rates $\eta$: SGD has prob. 0 of leaving $B_r$, and from any starting point there is a non-zero prob. for SGD to go in $B_r$.

## 1 INTRODUCTION

Several types of algorithmic bias have been observed in DNNs for a range of architectures (10; 34; 27; 28). Understanding and characterizing these types of implicit bias is crucial to understand the practical performances of Deep Neural Networks (DNNs).

We focus on Deep Linear Networks (DLNs) $A_\theta = W_L \cdots W_1$ for $\theta = (W_1, \ldots, W_L)$, that are known to be biased towards low-rank linear maps in a number of settings:

1. Adding $L_2$-regularization to the parameters of a DLN has the effect of adding $L_p$-Schatten norm (the $L_p$ norm of the singular values of a matrix) regularization to the learned matrix for $p = 2/L$ where $L$ is the depth of the network (9).

2. When trained with the cross-entropy loss, Gradient Descent (GD) diverges towards infinity along direction that maximizes the margin w.r.t. the parameter norm (17), leading to a form of implicit $L_2$-regularization with the same bias towards low-rank matrices.

3. When the parameters are initialized with a small variance, the network learns incrementally matrices of growing rank, thus converging to a low-rank solution (23; 14).

This low-rank bias is particularly useful in the context of matrix completion (5), where the goal is to recover a matrix from a subset of its entries under the assumption that the full matrix is low rank. The task of finding the lowest rank matrices fitting the observed entries is NP-hard, but convex approximations can work well (5; 6), as well as DLNs (18; 35).

In the deep case $L > 2$, the $L_p$-Schatten norm becomes non-convex (because $p = 2/L < 1$) and there are multiple local minima in the $L_2$-regularized loss, each corresponding to matrices with different ranks (similarly with cross-entropy there could be multiple directions that locally minimize the rank). Which of these local minima/critical point will GD converge to? And what will be the final rank?

We will see how Stochastic Gradient Descent (SGD) can lead the dynamics to jump between critical points with a bias towards low-rank points.

## 1.1 CONTRIBUTIONS

In this paper, we focus on the implicit bias of SGD in Deep Linear Networks (DLNs) of depth $L$ larger than 2 with $L_2$-regularization, when trained on Matrix Completion (MC) tasks.

We first describe the many critical points of the $L_2$-regularized loss. We then split them into three groups, depending on whether they recover the 'true rank', underestimate, or overestimate it.

We show that the rank-underestimating critical points can easily be avoided by taking a small enough ridge $\lambda$, but no such strategy exists to avoid rank-overestimating points with GD.

However we show SGD has a small but non-zero probability of jumping from any parameter to a lower rank parameter, but the probability of jumping to a higher rank is zero. More precisely, we define sets $B_r$ that contain all critical points of rank $r$ or less and show that they are absorbing: the probability for SGD to leave this set is zero, but the probability for SGD to move from outside of this set to inside (in sufficiently many steps) is non-zero.

This suggests that rank-overestimation can be avoided if we continue SGD training long enough (but not too long), since the rank will decrease incrementally. This illustrates the low-rank bias of SGD.

## 1.2 RELATED WORKS

The low-rank bias of DLNs has been observed in a number of different settings: for example as a result of $L_2$-regularization or training with the cross-entropy loss (9), and as a result of small initializations (2; 3; 23; 14). These results rely on similar tools such as the balancedness condition, however the underlying training dynamics leading to sparsity are very distinct.

Motivated by the empirical observation that SGD improves generalization (20; 19), there has been interest in the implicit bias of SGD. There is a line of work approximating SGD with different Stochastic Differential Equations (SDEs) (25; 33; 16; 7), sometimes approximating the parameter dependent noise covariance with a fixed scalar multiple of the identity thus leading to Langevin dynamics (16), and in general studying the resulting steady-state distributions (7). These SDE approximations require small learning rates (22), but approximations to capture the effect of large learning rates have been proposed too (22; 32).

These works however focus on the bias of SGD in parameter space, showing e.g. that it can be interpreted as changing the potential/loss (7), or adding a regularization term (32). More recent work has focused on the bias of SGD in diagonal linear networks (29; 36) leading to a sparsity effect in the vector represented by this network.

We focus on the effect of SGD in the context of deep fully-connected linear networks with $L_2$-regularization, showing that SGD strengthens the already existing low-rank bias induced by $L_2$-regularization. To our knowledge our work is also unique in that it does not rely on a SDE/continuous approximation.

## 2 SETUP

We study Deep Linear Networks (DLNs) of depth $L$ and widths $w_0 = d_{in}, w_1, \ldots, w_L = d_{out}$
$$A_\theta = W_L \cdots W_1,$$
for the $w_\ell \times w_{\ell-1}$ weight matrices $W_\ell$ and parameters $\theta = (W_1, \ldots, W_L)$. We will always assume that the number of neurons in the hidden layers is sufficiently large $w_\ell \geq \min\{d_{in}, d_{out}\}$ so that any $d_{out} \times d_{in}$ matrix $A$ can be recovered for some parameters $\theta$: $A = A_\theta$.

## 2.1 MATRIX COMPLETION

We consider the $L_2$-regularized loss
$$\mathcal{L}_\lambda(\theta) = C(A_\theta) + \lambda \|\theta\|^2$$
where $C$ is a loss on matrices such as the Matrix Completion (MC) loss
$$C(A) = \frac{1}{2N} \sum_{(i,j)\in I} \left(A^*_{ij} - A_{\theta,ij}\right)^2$$

where $A^*$ is the true matrix we want to recover and $I \subset \{1, \ldots, d_{out}\} \times \{1, \ldots, d_{in}\}$ is the set of observed entries of $A^*$ of size $N = |I|$. While it is not possible in general to recover an entire matrix $A^*$ from a subset of its entries, it is possible if $A^*$ is assumed to be low rank.

The ideal goal is to find the matrix $\hat{A}$ with lowest rank that matches the observed entries. We define the smallest rank as the smallest integer $r^*$ such that $\inf_{A:\text{Rank}A \leq r^*} C(A) = 0$. Note that one could also define $r^*$ to be the smallest integer where this infimum is attained at a finite matrix $A$, which can be higher in MC problems where filling in infinitely large entries can allow for lower ranks fitting functions. In the main we restrict ourselves to the first definition, but we discuss the second choice and its implications in Appendix A.

Since finding the minimal rank solution is NP-hard in general (5), a popular approximation is to find the matrix $\hat{A}$ that minimizes the MC loss with a nuclear norm regularization

$$\min_A \frac{1}{2N} \sum_{(i,j) \in I} \left(A_{ij}^* - A_{ij}\right)^2 + \lambda \|A\|_* ,$$

where the nuclear norm is the sum of the singular values of $A$: $\|A\|_* = \sum_{i=1}^{\text{Rank}A} s_i(A)$. This loss is convex and can be efficiently minimized, and it has been shown that it recovers the true matrix $A^*$ with high probability with an almost optimal number of observations (5; 6).

DLNs have also been used effectively in Matrix Completion, thanks to their implicit low-rank bias. The importance of low-rank bias in the Matrix Completion setting, makes it ideal to study the implicit bias of SGD in DLNs.

## 2.2 REPRESENTATION COST

The low-rank bias of $L_2$-regularized DLNs can be understood in terms of the representation cost $R(A; L)$ of DLNs, which equals the minimal parameter norm required to represent a matrix $A$ with a DLN of depth $L$:

$$R(A; L) = \min_{\theta : A = A_\theta} \|\theta\|^2 .$$

As observed in (9), the representation cost of DLNs equals the $L_p$-Schatten norm $\|A\|_p^p$ (the $L_p$ norm of the singular values) of $A$ for $p = 2/L$:

$$R(A; L) = L \|A\|_{2/L}^{2/L} := L \sum_{i=1}^{\text{Rank}A} s_i(A)^{2/L}.$$

This implies that the $L_2$-norm regularization in parameter space can be interpreted as adding a $L_p$-Schatten norm regularization in matrix space:

$$\min_\theta C(A_\theta) + \lambda \|\theta\|^2 = \min_A C(A) + \lambda L \|A\|_{2/L}^{2/L}.$$

For shallow networks ($L = 2$), the representation cost equals the nuclear norm $R(A; 2) = 2 \|A\|_*$. The loss has only global minima and strict saddles, thus guaranteeing convergence with probability 1 to a global minimizer $\hat{\theta}$ of the LHS, and the represented matrix $A_{\hat{\theta}}$ then minimizes the RHS. We therefore simply recover the convex relaxation of Matrix Completion, with the advantage that the loss $\mathcal{L}_\lambda(\theta)$ is differentiable everywhere, so that it can be optimized with vanilla GD (35).

In the deep ($L > 2$) case however, the representation cost $R(A; L) = L \|A\|_{2/L}^{2/L}$ is non-convex, and both RHS and LHS may have distinct local minima with varying rank. We will show that the correspondence extends to local minima of the RHS and LHS, and that there always exists one local minimum with the right rank $r^*$.

But there are multiple local minima and critical poinnts with different ranks, for example the zero parameters $\theta = 0$ corresponding to the zero matrix $A_\theta = 0$ is always a local minimum. Or there might also be critical points that overestimate the 'true rank' that we want to recover.

For GD with a Gaussian initialization, there is a non-zero probability to converge to any local minimum. On the other hand, we will see how SGD can jump from local minima to local minima.

## 2.3 STOCHASTIC GRADIENT DESCENT

We consider SGD with replacement, that is at each time step $t$ an index $(i_t, j_t)$ is sampled uniformly from the index set $I$, independently from the previous iterations. The parameters are then updated according to the learning rate $\eta$

$$\theta_{t+1} = (1 - 2\eta\lambda)\theta_t - \frac{\eta}{2}\nabla_\theta \left(A^*_{i_t j_t} - A_{\theta_t, i_t j_t}\right)^2.$$

Note that due to the $L_2$-regularization there remains noise even at the local minimizers, in contrast without $L_2$-regularization there is neither noise nor drift at the global minima of the loss. Thus with $L_2$-regularization the dynamics never completely stop, making it possible for SGD to jump from one local minimum to another.

**Remark 2.1.** *A number of previous works have approximated SGD by GD with Gaussian noise, the simplest of which is to approximate SGD by Langevin dynamics. Under this approximation, there is always a likelihood of jumping from local minimum to any other local minimum, with a higher likelihood of going to (and staying at) local minima with lower loss. Our theoretical results show a completely different behavior, where SGD may have non-zero probability of jumping from one local minimum to another, but zero likelihood of jumping back. Furthermore the likelihood of SGD visiting a certain local minimum will not scale with the loss of that local minimum, but rather its rank. This further shows that the Langevin approximation of SGD is inadequate.*

*It appears that what leads to this difference in dynamics, is that the covariance of the noise is low-dimensional and highly anisotropic, with a lot of noise along directions that keep or lower the rank, and almost no noise along directions that increase the rank.*

## 3 MAIN RESULTS

We will first give a description of the loss landscape of $L_2$-regularized DLNs and then state our main result, which says that SGD has a non-zero probability of jumping from parameters of a certain rank to a lower rank, and that once in the neighborhood of a low rank critical point, the probability of reaching a higher rank is zero.

### 3.1 $L_2$-REGULARIZED LOSS LANDSCAPE

The correspondence of the minimizers of $\mathcal{L}_\lambda(\theta)$ and $C_\lambda(A) := C(A) + \lambda L \|A\|_{2/L}^{2/L}$ extends to their local minima:

**Theorem 3.1.** *If $\hat{\theta}$ is a local minimum of $\mathcal{L}_\lambda(\theta)$, then $A_{\hat{\theta}}$ is a local minimum of $C_\lambda(A)$. Conversely, if $\hat{A}$ is a local minimum of $C_\lambda(A)$ then there is a local minimum $\hat{\theta}$ of $\mathcal{L}_\lambda(\theta)$ such that $\hat{A} = A_{\hat{\theta}}$.*

The critical points of the $L_2$-regularized loss are *balanced*, i.e. $W_\ell^T W_\ell = W_{\ell-1} W_{\ell-1}^T$ for all $\ell = 1, \dots, L-1$ (see Appendix A). This implies that all weight matrices have the same singular values and the same rank $r$. We may therefore define the rank of a critical point $\hat{\theta}$ as the rank $r$ of any weight matrix $W_\ell$ which also matches the rank of the represented matrix $A_{\hat{\theta}}$.

In general, there are several distinct local minima/critical points with different ranks. The origin $\theta = 0$ is always a local minimum (since the unregularized loss has vanishing first $L-1$ derivatives at 0, it becomes a local min even for any $\lambda > 0$), furthermore for small enough ridge $\lambda$, there always is a local minimum that finds the minimal rank required to fit the observed entries:

**Proposition 3.1.** *Consider a matrix completion problem with true matrix $A^*$ and observed entries $I$. For all $\lambda$, there is a rank $r^*$ local minima $\theta(\lambda)$ of $\mathcal{L}_\lambda(\theta)$ such that $\lim_{\lambda \searrow 0} C(A_{\theta(\lambda)}) = 0$.*

Note that finding a fitting matrix of minimal rank is known to be a NP-hard problem in general (5), which means that it should in general be hard to find this local minimum. There are three types of problematic local minima/critical points:

**Rank-underestimating critical points:** these are critical points such as the origin $\theta = 0$ with a rank lower than the minimal rank $r^*$, so that the represented matrix $A_\theta$ cannot fit the observed entries. GD with a small enough ridge $\lambda$ and learning rate $\eta$ will avoid such rank underestimating minima:

**Proposition 3.2.** *Given an initialization $\theta_0$ such that unregularized ($\lambda = 0$) gradient descent converges to a global minimum, there is a constant $c > 0$ such that for all small enough ridge $\lambda$ and learning rate $\eta$, regularized GD $\theta_t$ for all $t \geq t_0$ the $r^*$-th largest singular value of the matrix is non-zero: $s_{r^*}(A_{\theta_t}) \geq c$.*

There exists multiple papers proving convergence to a global minimum under different assumptions (1; 13; 26), but empirical evidence seem to suggest that convergence to a global minimum is much more general, and we expect more convergence results to come.

**Rank-overestimating critical points:** these have a larger rank than $r^*$ and the represented matrix $A_\theta$ can fit the observed entries (with a small $O(\lambda)$ error). These are harder to avoid, suggesting that the NP-hardness of finding an optimal rank $r^*$ fitting matrix can be related to avoiding these points. It might happen that there are no rank-overestimating minima/critical points, in which case GD can recover the minimal rank solution easily, but from now on we will focus on settings where these rank-overestimating points appear and how SGD manages to avoid them.

**'Bad' Rank-estimating local minima:** Finally there might exists local minima (or non-strict saddles where GD/SGD might also get stuck) with the right rank $r^*$ that do not fit the true matrix. This paper will not focus on how to avoid these points, but we believe that it could be proven that such minima are rare for large input and output dimensions, using similar tools as (4; 18). The noise of SGD might also help avoid these minima/saddles. In our numerical experiments, we never encountered such minima, but we did encounter both other types.

### 3.2 One-way Jumps from High to Low Rank

We now show how SGD helps avoiding rank-overestimating critical points. More precisely we show under conditions on the learning rate $\eta$ and ridge $\lambda$ that there is always a (small) likelihood of jumping to a lower rank, but the probability of jumping back is zero. This suggests a strategy: train the network with a small ridge to guarantee convergence to a critical point of at least the right rank, and then take advantage of the SGD noise to lower the rank until finding the right rank.

For our analysis, we define a family of regions $B_r \subset \mathbb{R}^P$ of parameters $\theta$ that are:

1. $\epsilon_1$-approximately balanced: for all layers $\ell$, $\left\| W_\ell^T W_\ell - W_{\ell-1} W_{\ell-1}^T \right\|_F^2 \leq \epsilon_1$,

2. $\epsilon_2, \alpha$-approximately rank $r$ (or less): for all $\ell$, $\sum_{i=1}^{\mathrm{Rank} W_\ell} f_\alpha(s_i(W_\ell^\top W_\ell)) \leq r + \epsilon_2$ where $s_i(A)$ is the $i$-th singular value of $A$ and $f_\alpha(x)$ is a twice differentiable function such that $f_\alpha(0) = 0$, $f_\alpha(x) = 1$ for $x > \alpha$, $0 \leq f'_\alpha(x) \leq \frac{K}{\alpha}$ for some constant $K$ and $f''_\alpha(x) \leq 0$.

3. $C$-bounded: $\|W_\ell\|_F^2 \leq C$.

Note that we chose the function $f_\alpha$ to be differentiable, and to satisfy $f_\alpha(0) = 0$ and $f_\alpha(x) = 1, \forall x \geq \alpha$. This yields a notion of approximate rank that converges to the true rank as $\alpha \searrow 0$. One example of such approximate rank is $f_\alpha(x) = \begin{cases} \dfrac{1}{\alpha^2} x(2\alpha - x), & x \leq \alpha \\ 1 & x > \alpha \end{cases}$.

Since all critical points $\hat{\theta}$ of the $L_2$-regularized loss are balanced, the set $B_r$ contains all critical points of rank $r$ or less for $C$ large enough and all $\epsilon_1, \epsilon_2 \geq 0$, and it contains no critical point of higher rank for $\epsilon_2$ and $\alpha$ small enough. These sets allows us to separate critical points by rank, with a small neighborhood.

**Proposition 3.3.** *For any critical point $\hat{\theta}$ in $B_r$, we have $\sum_{i=1}^{\mathrm{Rank} A_{\hat{\theta}}} f_\alpha(s_i(A_{\hat{\theta}})^{2/L}) \leq r + \epsilon_2$.*

*Proof.* Since $\hat{\theta}$ is balanced, $A_{\hat{\theta}} = U_L^\top S^L U_0$ where $S \in R^{d_{out} \times d_{in}}$ is the diagonal matrix of singular values of all $W_\ell$'s. Since for any $\ell$, $\sum_{i=1}^{\mathrm{Rank} W_\ell} f_\alpha(s_i(W_\ell^\top W_\ell)) \leq r + \epsilon_2$, $A_{\hat{\theta}}$ satisfies $\sum_{i=1}^{\mathrm{Rank} A_{\hat{\theta}}} f_\alpha(s_i(A_{\hat{\theta}}^{2/L})) \leq r + \epsilon_2$. □

We can now state our main result, which says that the set $B_r$ is absorbing for all $r$, i.e. SGD starting from anywhere will always end up at some time inside $B_r$ and then never leave it:

**Theorem 3.2.** *For any $r \geq 0$, $\lambda \geq 0$, any large enough $C$ and small enough $\epsilon_1, \epsilon_2, \alpha, \eta$, the set $B_r$ is closed*

$$\theta_t \in B_r \Rightarrow \theta_{t+1} \in B_r$$

*and for $r \geq 1$ and any parameters $\theta_t$ there is a time $T = \tilde{\Omega}(\lambda^{-1}\eta^{-1})$ (i.e up to log terms) such that*

$$\mathbb{P}\left(\theta_{t+T} \in B_r | \theta_t\right) \geq \left(\frac{r}{\min\{d_{in}, d_{out}\}}\right)^T,$$

*thus for any starting point SGD will eventually reach $B_r$:*

$$\mathbb{P}(\exists T : \theta_{t+T} \in B_r | \theta_t) = 1.$$

*Proof.* (sketch) (1) The closeness of the set of $\epsilon_1$-approximately balanced parameters follows from the fact that in the gradient flow limit $\eta \searrow 0$, the balancedness errors $W_\ell^T W_\ell - W_{\ell-1} W_{\ell-1}^T$ decay exponentially

$$\partial_t \left(W_\ell^T W_\ell - W_{\ell-1} W_{\ell-1}^T\right) = -\lambda \left(W_\ell^T W_\ell - W_{\ell-1} W_{\ell-1}^T\right).$$

To guarantee a similar decay with SGD, we simply need to the control the $O(\eta^2)$ terms.

Given $\epsilon_1$-approximately balancedness, the closeness of the $\epsilon_2$-approximately rank $r$ or less parameters follows from the fact that the dynamics resulting from the minimization of the cost $C(A_\theta)$ are very slow along the smallest singular vectors of $A_\theta$ (1) but the $L_2$-regularization term pushes these small singular values towards zero. For small enough singular values, this second force dominates, thus leading to a decay towards zero.

(2) Under the event $A_T$ that in the steps $s$ from $t$ to $t + T - 1$ all the random entries $(i_s, j_s)$ are sampled from the same $r$ of the $d_{out}$ columns, one can show that the $d_{out} - r$ other columns of $W_L$ decay exponentially to approximately 0, implying an approximate rank of $r$ or less. The probability of that event is at least $\left(\frac{r}{d_{out}}\right)^T$. $\square$

This shows the implicit bias of SGD towards low-rank matrices in matrix completion: SGD can avoid any rank-overestimating minima/critical point given enough training steps.

Explicit bounds on $C, \alpha, \epsilon_1, \epsilon_2, \eta$ can be found in Appendix B. The bounds are rather complex, but we give here an example of acceptable rates in terms of $\lambda$: $C \sim \lambda^{-1}, \alpha \sim \lambda^{\frac{L+2}{L-2}}, \epsilon_1 \sim \lambda^{2\frac{L+2}{L-2}+2L+1}$, $\epsilon_2 \sim \lambda^0$, and $\eta \sim \lambda^{4L+1+\frac{L+1}{L-2}}$. These rate suggest that an extremely small learning rate $\eta$ is necessary, especially for large depths $L$, thus making the likelihood of a jump appear very small. This seems in contradiction with our empirical observations that larger depths tend to make these jumps more likely. We believe our bounds could be made tighter, in particular when it comes to the dependence on the depth $L$ to better reflect our empirical observations.

We expect this result to generalize to other tasks. The first part of the theorem (the closeness of $B_r$) should generalize to costs such as the MSE loss and others, and the second part too, under the event that one samples from the same $r$ training points over $T$ time steps for the MSE loss, or sample from the same $r$ classes for classification tasks.

A limitation however is that the second part of the result relies on the fact that we sample the observed entries independently at each time $t$ with possible replacement. In practice, the dataset is randomly shuffled and taken in this random order, so that every observed entry is chosen exactly once during each epoch. This would force the jumps to happen within an epoch, which may not be possible depending on the problem.

Another limitation is the average time required to observe one of our predicted jumps can easily be absurdly large. To observe a jump in reasonable time, one also needs rather large learning rates $\eta$, leading to very noisy dynamics. This makes periodic learning rate choices attractive, with large $\eta$ periods allowing for jumps to lower-rank region, and low $\eta$ periods allowing for SGD to settle around a local minimum.

Nevertheless, our result also shows that the common approach of approximating SGD with a SDE such as Langevin dynamics and studying the stationary distribution (usually with full support over the parameter space) is misleading. In contrast, our result implies that any stationary distribution must have support inside $B_{r=1}$, thus under-estimating the true rank in general. It is thus crucial to understand the distribution of SGD at intermediate times, when the rank has not yet collapsed to 1.

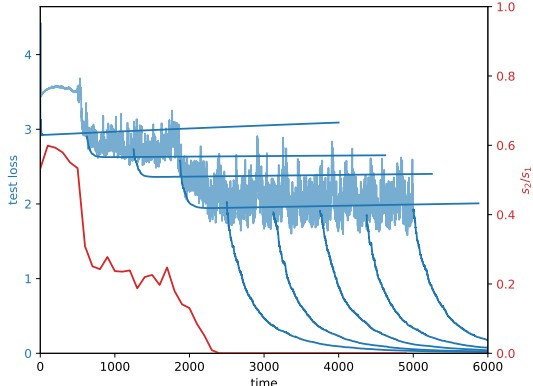 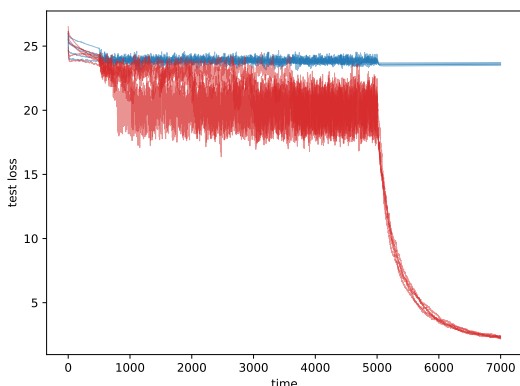

Figure 1: **Annealing Schedule:** DLN with $L = 3$, $w_1 = 100$ on the $2 \times 2$ MC problem with $\epsilon = 0.25$ [Light blue] SGD with $\lambda = 0.1$ and $\eta = 0.2$ ($\eta = 0.03$ for the first 500 steps to avoid explosion) [dark blue] at different times, we create offshoots with $\lambda = 0.001$ and $\eta = 0.02$ to fit the data. [red] The ratio of the second to first singular value of $A_\theta$ on the large $\lambda, \eta$ path. We see a jump around time 2000, where the output matrix becomes rank 1. The offshoots created before this jump fail to fit the missing entry while those created after succeed.

Figure 2: **Effect of depth:** We study the effect of on the MC problem with $\epsilon = 0.1$. We train 5 networks of each depths $L = 3$ [blue] and $L = 4$ [red] with the same schedule: $\eta = 0.03, \lambda = 0.1$ until $t = 500$, then $\eta = 0.25, \lambda = 0.1$ until $t = 5000$ and finally $\eta = 0.05, \lambda = 0.001$ until the end. We see that the five depth $L = 3$ networks are unable to jump in this time, while all five depth $L = 4$ networks jump at different times during the first 5000 SGD steps.

### 3.2.1 NONLINEAR NETWORKS

Since linear networks are a simplification of nonlinear networks, it is natural to wonder whether the results presented here could be generalized to the nonlinear case. We identify two possible strategy to generalize our results:

First along the lines of (8) which observes a similar phenomenon where SGD is naturally attracted to symmetric regions of the loss (where for example two neurons are identical or one neuron is dead) in nonlinear networks. The $L_2$-regularization is known to make these region more attractive (15), which could have a compound effect with SGD. In DLNs, the regions of low rank that we prove are attractive can also be interpreted as neighborhoods of symmetric / invariant regions.

Second, recent work has shown that $L_2$ regularized ReLU DNNs with large depths are biased towards minimizing a notion of rank over nonlinear functions, the Bottleneck rank (12). We have hope that our results could be extended to prove a similar low-rank bias with this new notion of rank. This is further motivated by the observation that such large depth networks exhibit a Bottleneck structure (11) where the middle layers of the network behave approximately like linear layers.

## 4 NUMERICAL EXPERIMENTS

For our numerical experiments, we want to find a Matrix Completion problem that GD cannot solve but SGD can. In particular, we want to find a setup where GD converges with a high probability to a rank-overestimating minimum, and where SGD can jump from this minimum to a lower rank minimum in a reasonable amount of time.

It is rather difficult to find a setup that lies between the regimes where both GD and SGD work and where neither work. This is in line with previous work in the bias of SGD (29): in diagonal networks a value (determined by the initialization) determines a transition between a sparse and non-sparse regimes, and SGD has the effect of pushing this value towards the sparse regime; this can have a significant sparsity effect if the original value was at the transition between regimes, but little effect if it was far into either regimes.

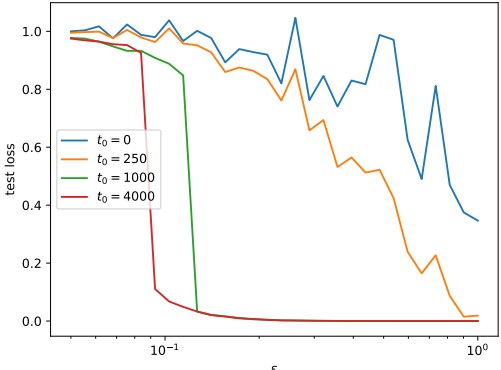 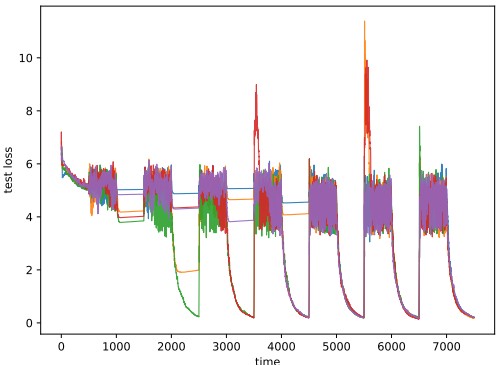

Figure 3: **Annealing accross $\epsilon$:** For a range of $\epsilon$, we train 4 networks ($L = 4$, $w = 100$) with an annealing schedule as in Figure 1 and plot the test loss divided by the test loss when putting zeros in the missing entries. The four networks are trained for $t_0$ steps with high noise, followed by 4000 steps in the low noise regime. Without a noise phase ($t_0 = 0$) the network fails to recover the rank 1 solution. Larger $t_0$ allow the network to recover it for even smaller $\epsilon$.

Figure 4: **Periodic Schedule:** DLN with $L = 3$, $w_1 = 100$ on the $2 \times 2$ MC problem with $\epsilon = 0.2$. We plot 5 runs of SGD with periodic learning rates alternating between $\eta = 0.1, \lambda = 0.001$ and $\eta = 0.4, \lambda = 0.1$. We see that the different trials make jumps during the high $\eta, \lambda$ period. After the jump, SGD will settle at a low test error in the slow periods, allowing us to identify when the jump happened.

We choose a MC problem, inspired by (30), that allows us to tune the difficulty of finding a sparse solution. We observe 3 out of 4 entries of a $2 \times 2$ matrix:

$$\begin{pmatrix} 1 & * \\ \epsilon & 1 \end{pmatrix}.$$

Filling the missing entry $*$ with $\epsilon^{-1}$ leads to a rank 1 matrix. The smaller $\epsilon$ is, the larger the missing entry that needs to be filled in needs to be.

In $L_2$-regularized DLNs with $L > 2$ there are three local minima: the rank 0 minimum at the origin which can easily be avoided, a set of minima that learn the rank 1 solution, and a set of rank-overestimating minima that learn a rank 2 solution by filling the missing entry with a small value.

For small $\epsilon$ values, GD almost always converges to a rank-overestimating minimum (see Figure 3). In such setup, SGD can outperform GD by jumping to a rank 1 solution. To achieve a jump in a reasonable amount of time, we need the ridge parameter $\lambda$ and the learning rate $\eta$ to be large. But such a choice of large $\lambda, \eta$ prevent SGD from minimizing the train error.

We investigate two strategies to take advantage of both the jumping properties of large $\lambda, \eta$ and fitting properties of small $\lambda, \eta$:

**'Annealing' Schedule:** In Figure 1, we run SGD with large $\lambda, \eta$ for some time $t_0$, waiting for a jump and then switch to small values of $\lambda, \eta$ for convergence. Another specificity is that we take a small learning rate for the first few steps, because SGD diverges if we start with a too large learning rate directly at initialization, whereas large learning rates are possible after a few steps (we do not have a theoretical explanation for that).

We test different switching times to small $\lambda, \eta$ values, and we see clearly that if we switch after the jump at time $\sim 2000$, we obtain a rank 1 solution, but if we switch before the jump then training fails and recovers a rank 2 solution.

By changing $\epsilon$ we can tune the difficulty of finding the rank 1 solution. We see in Figure 3 that the smaller $\epsilon$, the longer one needs to wait for a jump, and thus the longer one needs to stay in the high noise setting. We also see that without a high noise period (i.e. when we are close to GD) the network fails to recover the rank 1 solution even for $\epsilon = 1$.

**Periodic Schedule:** Another strategy it to alternate between large and small $\lambda, \eta$. We see in Figure 4 how the jumps all happen during the large $\lambda, \eta$ periods. It is also interesting to see that even after SGD has settled in the vicinity of a local minimum in one of the small $\lambda, \eta$ periods, SGD can still jump to another minimum in a subsequent large $\lambda, \eta$ period.

Finally we also study the effect of depth in Figure 2, and observe that depth increases the probability of jumps. We train networks of depths $L = 3$ and $L = 4$ on the $2 \times 2$ MC task with $\epsilon = 0.1$. While for the choice $\epsilon = 0.25$, a depth $L = 3$ network was able to jump in a reasonable amount of time, for this smaller choice of $\epsilon$ we do not observe a jump (even with the same hyper-parameters). In contrast, the deeper networks $L = 4$ all jump in a reasonable amount of time, suggesting that depth increases the likelihood of a jump.

## 5 CONCLUSION

We have given a description of the loss landscape of $L_2$-regularized DLNs, giving a classification of its critical points by their rank. We have then shown that SGD has a non-zero probability of jumping from any higher rank critical point to a lower rank one, but it has a zero probability of jumping in the other direction. We observe these jumps empirically. To our knowledge, this is the first description of the low-rank bias of SGD in the context of fully-connected linear networks with two or more hidden layers.

Our analysis is also significantly different from previous approaches that rely on approximating SGD with a continuous stochastic process, and/or studying of the limiting distribution of this continuous process. It appears that the phenomenon of absorbing sets of different ranks cannot be recovered with a continuous approximation, and the jumps we describe happen before SGD has reached its limiting distribution. This puts into question the adequacy of the continuous approximation and limiting distribution assumption.

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

The Appendix is organized as follows:

- Section A contains the proofs of Theorem 3.1 and Propositions 3.1 and 3.2 of the main.
- Section B then describes how Theorem 3.2 of the main can be split into two statements: Theorems B.1 and B.2.
- Section C state some preliminary result for the proofs.
- Section D proves the first part of Theorem 3.2 from the main, that is Theorem B.1.
- Section E proves the second part of Theorem 3.2, that is Theorem B.2.
- Section F states and proves a more general version of Proposition 3.3 of the main.

## A  LOSS LANDSCAPE

**Proposition A.1.** *Let $\hat{\theta}$ be a critical point of the loss $\mathcal{L}_\lambda$, then $\hat{\theta}$ is balanced, i.e. $W_\ell W_\ell^T = W_{\ell+1}^T W_{\ell+1}$.*

*Proof.* At a critical point, we have

$$W_{\ell+1}^T \cdots W_L^T \nabla C(A_\theta) W_1^T \cdots W_{\ell-1}^T + 2\lambda W_\ell = 0.$$

Thus

$$W_\ell W_\ell^T = -\frac{1}{2\lambda} W_{\ell+1}^T \cdots W_L^T \nabla C(A_\theta) W_1^T \cdots W_\ell^T = W_{\ell+1}^T W_{\ell+1}.$$

$\square$

**Proposition A.2.** *Let $\hat{\theta}$ be a critical point of the loss $\mathcal{L}_\lambda$, then:*

- *$\hat{\theta}$ is a local minimum if and only if $A_{\hat{\theta}}$ is a local minimum of $C_\lambda$.*

- *$\hat{\theta}$ is a strict saddle/maximum if and only if $A_{\hat{\theta}}$ is a strict saddle/maximum.*

*Proof.* We know that any critical point of the $L_2$-regularized loss is balanced. The parameters $\hat{\theta} = (W_1, \ldots, W_L)$ are therefore of the form

$$W_\ell = U_\ell S^{\frac{1}{L}} U_{\ell-1}^T,$$

for some $d \times d$ diagonal $S$ (where $d = \min\{d_{in}, d_{out}\}$) and $w_\ell \times d$ matrices $U_\ell$ with orthonormal columns ($U_\ell^T U_\ell = I_d$).

(0) For any sequence of matrices $A_1, A_2, \ldots$ converging to $A_{\hat{\theta}}$ with SVD decompositions $A_i = \tilde{U}_i S_i \tilde{V}_i^T$ (chosen so that $\tilde{U}_i$, $S_i$ and $\tilde{V}_i$ converge to the SVD decomposition $A_{\hat{\theta}} = U_L S U_0^T$) we can construct parameters $\theta_i$ with weight matrices

$$W_1 = U_1 S_i^{\frac{1}{L}} \tilde{V}_i^T$$
$$W_\ell = U_\ell S_i^{\frac{1}{L}} U_{\ell-1}^T$$
$$W_L = \tilde{U}_i S_i^{\frac{1}{L}} U_{L-1}^T.$$

We have (1) $A_{\theta_i} = A_i$, (2) $\theta_i \to \hat{\theta}$, (3) $\|\theta_i\|^2 = L \|A_i\|_{2/L}^{2/L}$ and therefore $C_\lambda(A_i) = \mathcal{L}_\lambda(\theta_i)$.

(1a) If $\hat{\theta}$ is not a local minimum, there is a sequence $\theta_i \to \hat{\theta}$ with $\mathcal{L}_\lambda(\theta_i) < \mathcal{L}_\lambda(\hat{\theta})$, thus the sequence $A_i = A_{\theta_i}$ converges to $A_{\hat{\theta}}$ and $C_\lambda(A_i) \le \mathcal{L}_\lambda(\theta_i) < \mathcal{L}_\lambda(\hat{\theta}) = C_\lambda(A_{\hat{\theta}})$, implying that $A_{\hat{\theta}}$ is not a local minimum.

(1b) If $A_{\hat{\theta}}$ is not a local minimum, there is a sequence $A_i \to A_{\hat{\theta}}$ with $C_\lambda(A_i) < C_\lambda(A_{\hat{\theta}})$, by point (0), we construct a sequence $\theta_i \to \hat{\theta}$ such that $\mathcal{L}_\lambda(\theta_i) = C_\lambda(A_i) < C_\lambda(A_{\hat{\theta}}) = \mathcal{L}_\lambda(\hat{\theta})$, proving that $\hat{\theta}$ is not a local minimum.

(2a) $\hat{\theta}$ is a strict saddle/maximum if there is a sequence $\theta_i \to \hat{\theta}$ such that

$$\lim_{i \to \infty} \frac{\mathcal{L}_\lambda(\theta_i) - \mathcal{L}_\lambda(\hat{\theta})}{\|\theta_i - \hat{\theta}\|^2} < 0.$$

Our goal is now to show that the sequence $A_{\theta_i}$ that converges to $A_{\hat{\theta}}$ satisfies the same property. First note that

$$\lim_{i \to \infty} \frac{C_\lambda(A_{\theta_i}) - C_\lambda(A_{\hat{\theta}})}{\left\|A_{\theta_i} - A_{\hat{\theta}}\right\|_F^2} \leq \lim_{i \to \infty} \frac{\mathcal{L}_\lambda(\theta_i) - \mathcal{L}_\lambda(\hat{\theta})}{\|\theta_i - \hat{\theta}\|_F^2} \frac{\|\theta_i - \hat{\theta}\|_F^2}{\left\|A_{\theta_i} - A_{\hat{\theta}}\right\|_F^2},$$

since $C_\lambda(A_{\theta_i}) \leq \mathcal{L}_\lambda(\theta_i)$ and $C_\lambda(A_{\hat{\theta}}) = \mathcal{L}_\lambda(\hat{\theta})$.

Since $A_\theta$ is twice differentiable w.r.t. $\theta$, we know that $\left\|A_{\theta_i} - A_{\hat{\theta}}\right\|_F \leq c\|\theta_i - \hat{\theta}\| + O\left(\|\theta_i - \hat{\theta}\|^2\right)$ and thus

$$\frac{\|\theta_i - \hat{\theta}\|^2}{\|A_{\theta_i} - A_{\hat{\theta}}\|_F^2} \geq \frac{\|\theta_i - \hat{\theta}\|^2}{\left(c\|\theta_i - \hat{\theta}\| + O\left(\|\theta_i - \hat{\theta}\|^2\right)\right)^2} = \frac{1}{\left(c + O\left(\|\theta_i - \hat{\theta}\|\right)\right)^2} \geq \frac{1}{4c^2},$$

for large enough $i$. This implies that

$$\lim_{i \to \infty} \frac{C_\lambda(A_{\theta_i}) - C_\lambda(A_{\hat{\theta}})}{\left\|A_{\theta_i} - A_{\hat{\theta}}\right\|_F^2} \leq \lim_{i \to \infty} \frac{\mathcal{L}_\lambda(\theta_i) - \mathcal{L}_\lambda(\hat{\theta})}{\|\theta_i - \hat{\theta}\|_F^2} \frac{1}{c^2} < 0,$$

where we used the fact that for $i$ large enough $\frac{\mathcal{L}_\lambda(\theta_i) - \mathcal{L}_\lambda(\hat{\theta})}{\|\theta_i - \hat{\theta}\|_F^2}$ is negative.

This implies that $A_{\hat{\theta}}$ is a strict saddle/maximum.

(2b) $A_{\hat{\theta}}$ is a strict saddle if there is a sequence $A_i \to A_{\hat{\theta}}$ with

$$\lim_{i \to \infty} \frac{C_\lambda(A_i) - C_\lambda(A_{\hat{\theta}})}{\left\|A_i - A_{\hat{\theta}}\right\|_F^2} < 0.$$

For $L > 2$ we may assume that $A_i$ has the same rank as $A_{\hat{\theta}}$ for large enough $i$: a matrix cannot be approached with matrices of strictly lower rank (the best rank $k$ approximation matches the $k$ largest singular values leading to an error larger than the $k+1$ largest singular value), and if it is approached with a strictly larger rank the regularization term $\|A_i\|_{2/L}^{2/L}$ would be strictly larger.

We now construct a sequence $\theta_i \to \hat{\theta}$ as in (0). Consider the map $\phi$ that maps matrices $A = \tilde{U}S\tilde{V}^T$ in the neighborhood of $A_{\hat{\theta}}$ with the same rank to the parameters

$$W_1 = U_1 S^{\frac{1}{L}} \tilde{V}^T$$
$$W_\ell = U_\ell S^{\frac{1}{L}} U_{\ell-1}^T$$
$$W_L = \tilde{U} S^{\frac{1}{L}} U_{L-1}^T.$$

We have $\phi(A_i) = \theta_i$ and $\phi(A_{\hat{\theta}}) = \hat{\theta}$. And since $\phi$ is differentiable at $A_{\hat{\theta}}$ along directions that do not change the rank (37), we have

$$\lim_{i \to \infty} \frac{\mathcal{L}_\lambda(\theta_i) - \mathcal{L}_\lambda(\hat{\theta})}{\left\|\theta_i - \hat{\theta}\right\|^2} = \lim_{i \to \infty} \frac{C_\lambda(A_i) - C_\lambda(A_{\hat{\theta}})}{\left\|A_i - A_{\hat{\theta}}\right\|_F^2} \frac{\left\|A_i - A_{\hat{\theta}}\right\|_F^2}{\left\|\phi(A_i) - \phi(A_{\hat{\theta}})\right\|^2} < 0.$$

$\square$

Finally we prove the existence of a minimum with the minimal rank required to fit the data:

**Proposition A.3.** *Consider a matrix completion problem with true matrix $A^*$ and observed entries $I$. For all $\lambda$, there is a rank $r^*$ local minima $\theta(\lambda)$ of $\mathcal{L}_\lambda(\theta)$ such that $\lim_{\lambda \searrow 0} C(A_{\theta(\lambda)}) = 0$.*

*Proof.* For all $\lambda$, take the matrix $A(\lambda)$ to minimize the cost $C_\lambda(A)$ amongst the set of matrices of rank $r^*$ or less. The regularization ensures that the infimum $\inf_{A:\mathrm{Rank}A \le r^*} C_\lambda(A)$ is attained at a finite matrix $A$.

The matrix $A(\lambda)$ is also a (possibly non-global) minimum of the non-restricted loss: it is locally optimal amongst matrices of the same rank, and for any matrix $A'$ of rank $r > r^*$, define $A'_{\le r^*}$ to be the closest rank $r^*$ matrix (which is obtained by setting all but the $r^*$-th largest singular values of $A'$ to zero), we have

$$C_\lambda(A') = C_\lambda(A'_{\le r^*}) + \lambda L \|A' - A'_{\le r^*}\|_{2/L}^{2/L} + [C(A') - C(A'_{\le r^*})].$$

We have that $C_\lambda(A'_{\le r^*}) \ge C_\lambda(A(\lambda))$ and in a small enough neighborhood of $A(\lambda)$, one can guarantee that the second term dominates the third one since $C$ is differentiable (if $s_{r^*+1} > 0$ is the $r^*+1$-th largest singular value of $A'$, then the second term is at least $\lambda L s_{r^*+1}^{2/L}$, while the second one is at most $c(\mathrm{Rank}A' - r^*)s_{r^*+1}$ where $c < \infty$ bounds the Lipschitzness of $C$ in the neighborhood).

By Proposition A.2 there are corresponding local minima $\theta(\lambda)$ of $\mathcal{L}_\lambda(\theta)$ such that $A(\lambda) = A_{\theta(\lambda)}$. By the definition of the minimal rank $r^*$, we know that $\inf_{A:\mathrm{Rank}A \le r^*} C_\lambda(A) = 0$ and thus $\lim_{\lambda \searrow 0} C(A_{\theta(\lambda)}) = 0$. $\qquad\square$

## A.1 Avoiding Rank-underestimating Critical Points

With a small enough ridge $\lambda$ and learning rate $\eta$, one can guarantee that GD will avoid all rank-underestimating local minima:

**Proposition A.4** (Proposition 3.2 in the main). *Given an initialization $\theta_0$ such that unregularized ($\lambda = 0$) gradient descent converges to a global minimum, there is a constant $c > 0$ such that for all small enough ridge $\lambda$ and learning rate $\eta$, regularized GD $\theta_t$ for all $t \ge t_0$ the $r^*$-th largest singular value of the matrix is non-zero: $s_{r^*}(A_{\theta_t}) \ge c$.*

*Proof.* Since unregularized GD converges to a global minimum with zero loss, there is a time $t_0$ such that $\mathcal{L}(\theta_{t_0,\lambda=0}) \le \frac{1}{4} \inf_{A:\mathrm{Rank}A < r^*} C(A)$.

We know that $L_2$-regularized GD $\theta_\lambda(t)$ converges to unregularized GD $\theta(t)$ as $\lambda \searrow 0$ for any fixed time $t$. Then for all sufficiently small $\lambda$ we have $\mathcal{L}_\lambda(\theta_{t_0,\lambda}) \le \frac{1}{2} \inf_{A:\mathrm{Rank}A < r^*} C(A)$.

Lemma A.5 also tells us that GD is non-increasing for a small enough $\eta$, so that for all $t \ge t_0$ we also have

$$\mathcal{L}_\lambda(\theta_{t,\lambda,\eta}) \le \frac{1}{2} \inf_{A:\mathrm{Rank}A < r^*} C(A).$$

Let us now show that there is a constant $c > 0$ such that the $r^*$-th largest singular value $s_{r^*}(A_{\theta_{t,\lambda,\eta}})$ of $A_{\theta_{t,\lambda,\eta}}$ is lower bounded by $c$.

Assume by contradiction that this is not the case, then for all $\epsilon$, there is a matrix $A_\epsilon$ with $s_r(A_\epsilon) \le \epsilon$ for all $r \ge r^*$ and with $C_\lambda(A_\epsilon) \le \frac{1}{2} \inf_{A:\mathrm{Rank}A < r^*} C(A)$.

We know that the matrices $A_\epsilon$ are bounded for all $\epsilon$ since $\|A_\epsilon\|_{2/L}^{2/L} \le \frac{1}{\lambda} C_\lambda(A_\epsilon) \le \frac{1}{2\lambda} \inf_{A:\mathrm{Rank}A < r^*} C(A) < \infty$. There is thus a convergent sequence $A_{\epsilon_1}, A_{\epsilon_2}, \dots$ with $\epsilon_n \to 0$ as $n \to \infty$, whose limit $\hat{A} = \lim_{n\to\infty} A_{\epsilon_n}$ is finite, has rank strictly less than $r^*$ and satisfies $C(\hat{A}) \le C_\lambda(\hat{A}) \le \frac{1}{2} \inf_{A:\mathrm{Rank}A < r^*} C(A)$ which yields a contradiction with the fact that $\inf_{A:\mathrm{Rank}A < r^*} C(A) > 0$. $\qquad\square$

Here are a few Lemmas required for the previous proof:

**Lemma A.1.** *The loss gradient $\nabla \mathcal{L}_\lambda(\theta)$ in the ball $B(0, D)$ is Lipschitz with constant $H = \frac{2L\sqrt{(L-1)}}{N}(2D^{2L-2} + cD^{L-2} + 2\lambda)$.*

*Proof.* For all parameters $\theta$ with $\|\theta\| \le D$, we have $\|W_\ell\|_F \le D$ and thus $\|A_\theta\|_F \le D^L$. Furthermore for any two parameters $\theta = (W_1, \dots, W_L)$ and $\phi = (V_1, \dots, V_L)$ with $\|\theta\|, \|\phi\| \le D$, we

have

$$\|A_\theta - A_\phi\|_F \le \|W_L \cdots W_2(W_1 - V_1)\|_F + \cdots + \|(W_L - V_L)V_{L-1}\cdots V_1\|_F$$

$$\le D^{L-1} \sum_{\ell=1}^{L} \|W_\ell - V_\ell\|_F$$

$$\le \sqrt{L}D^{L-1} \|\theta - \phi\|.$$

Now since $\nabla C(A_\theta) = \frac{2}{N} M \odot (A_\theta - A^*)$, this implies that

$$\|\nabla C(A_\theta)\|_F \le \frac{2}{N} \|A_\theta\|_F + \frac{2}{N} \|A^*\|_F \le \frac{2}{N} D^L + c$$

and

$$\|\nabla C(A_\theta) - \nabla C(A_\phi)\|_F \le \frac{2}{N} \|A_\theta - A_\phi\|_F \le \frac{2}{N} \sqrt{L}D^{L-1} \|\theta - \phi\|.$$

Now using these facts, we get that the difference $\|\nabla \mathcal{L}_\lambda(\theta) - \nabla \mathcal{L}_\lambda(\phi)\|^2$ is upper bounded by

$$\sum_{\ell=1}^{L} \left\| W_{\ell+1}^T \cdots W_L^T \nabla C(A_\theta)W_1^T \cdots W_{\ell-1}^T - V_{\ell+1}^T \cdots V_L^T \nabla C(A_\phi)V_1^T \cdots V_{\ell-1}^T + 2\lambda W_\ell - 2\lambda V_\ell \right\|^2$$

$$\le \sum_{\ell=1}^{L} \sum_{k\ne\ell} \left( D^{L-2} \|W_k - V_k\|_F \|\nabla C(A_\theta)\| + D^{L-1} \|\nabla C(A_\theta) - \nabla C(A_\phi)\| + 2\lambda \|W_\ell - V_\ell\|_F \right)^2$$

$$\le \sum_{\ell=1}^{L} \sum_{k\ne\ell} \left( \frac{2}{N}(D^{2L-2} + cD^{L-2}) \|W_k - V_k\|_F + \frac{2}{N}\sqrt{L}D^{2L-2} \|\theta - \phi\| + 2\lambda \|W_\ell - V_\ell\|_F \right)^2$$

$$\le \sum_{\ell=1}^{L} \sum_{k\ne\ell} \left( \frac{2}{N}\sqrt{L}(2D^{2L-2} + cD^{L-2} + 2\lambda) \|\theta - \phi\| \right)^2$$

$$= \frac{4L^2(L-1)}{N^2}(2D^{2L-2} + cD^{L-2} + 2\lambda)^2 \|\theta - \phi\|^2.$$

and thus

$$\|\nabla \mathcal{L}_\lambda(\theta) - \nabla \mathcal{L}_\lambda(\phi)\| \le \frac{2L\sqrt{(L-1)}}{N}(2D^{2L-2} + cD^{L-2} + 2\lambda) \|\theta - \phi\|.$$

$$\square$$

**Proposition A.5.** *Given initial parameters $\theta_0$, we have that GD with ridge $\lambda$ and learning rate $\eta < \frac{2}{H}$ for $D = \sqrt{\frac{\mathcal{L}_\lambda(\theta_0)}{\lambda}}$ and $H = \frac{2L\sqrt{(L-1)}}{N}(2D^{2L-2} + cD^{L-2} + 2\lambda)$ remains inside the ball $B(0, D)$ and has a non-increasing regularized loss.*

*Proof.* Given the parameters at any time step $\theta_t$ in the ball $B(0, D)$ the loss along the gradient $\theta_t - \eta\nabla\mathcal{L}_\lambda(\theta_t)$ (for any $\eta$ small enough so that this next step remains inside the ball $B(0, D)$) satisfies

$$\mathcal{L}_\lambda(\theta_t - \eta\nabla\mathcal{L}_\lambda(\theta_t)) - \mathcal{L}_\lambda(\theta_t) = \int_0^\eta \nabla\mathcal{L}_\lambda(\theta_t)^T \nabla\mathcal{L}_\lambda(\theta_t - p\nabla\mathcal{L}_\lambda(\theta_t))dp$$

$$\le -\eta \|\nabla\mathcal{L}_\lambda(\theta_t)\|^2 + \eta \int_0^\eta \|\nabla\mathcal{L}_\lambda(\theta_t)\| \|\nabla\mathcal{L}_\lambda(\theta_t) - \nabla\mathcal{L}_\lambda(\theta_t - p\nabla\mathcal{L}_\lambda(\theta_t))\| \, dp$$

$$\le -\eta \|\nabla\mathcal{L}_\lambda(\theta_t)\|^2 + \frac{\eta^2}{2}H \|\nabla\mathcal{L}_\lambda(\theta_t)\|^2$$

for $H = \frac{2L\sqrt{(L-1)}}{N}(2D^{2L-2} + cD^{L-2} + 2\lambda)$. Thus for any $\eta \le \frac{2}{H}$, the loss is non-increasing:

$$\mathcal{L}_\lambda(\theta_t - \eta\nabla\mathcal{L}_\lambda(\theta_t)) - \mathcal{L}_\lambda(\theta_t) \le 0.$$

Choosing $D = \sqrt{\frac{\mathcal{L}_\lambda(\theta_0)}{\lambda}}$, we know that $\theta_0 \in B(0, D)$ since $\|\theta_0\|^2 \leq \frac{\mathcal{L}_\lambda(\theta_0)}{\lambda}$ and all subsequent steps $\theta_t$ with a learning rate $\eta < \frac{2}{H}$ will remain inside the ball, since $\mathcal{L}_\lambda(\theta_0)$ is non-increasing along the continuous path made up of the concatenation of the segments $[\theta_t, \theta_{t+1}]$ it can never leave the ball since $\|\theta_t\|^2 \leq \frac{\mathcal{L}_\lambda(\theta_t)}{\lambda} \leq \frac{\mathcal{L}_\lambda(\theta_0)}{\lambda}$ (where $t$ is a real value that can lie on any segment). $\qquad\square$

### A.1.1 DIFFERENT NOTION OF OPTIMAL RANK

In settings where the infimum $\inf_{A:\text{Rank}A \leq r^*} C(A)$ is not attained, we can define another notion of smallest rank $\tilde{r}^* > r^*$ to be the smallest integer where this infimum is attained. One could wonder under which conditions one can avoid minima with rank $< \tilde{r}^*$. A similar result can be proven, though we require an additional assumption (and for simplicity, it is proven for GF instead of GD):

**Proposition A.6.** *Given an initialization $\theta_0$ such that unregularized ($\lambda = 0$) gradient flow (GF) converges to a global minimum $\theta_\infty$ such that the loss is $\beta$-PL in a neighborhood of $\theta_\infty$, then for $\lambda$ small enough, regularized GF converges to a minimum rank no smaller than $\tilde{r}^*$.*

*Proof.* If we let the ridge $\lambda$ go to zero we have that GF trained with $\lambda$-weight decay $\theta_{t,\lambda}$ converges to GF without weight decay: $\theta_{t,\lambda} \to \theta_t$ as $\lambda \searrow 0$. We can therefore choose a time $t_0$ large enough and small enough $\lambda_0$ such that for all $\lambda \leq \lambda_0$, the ball $B(\theta_{t_0,\lambda}, R)$ for $R = \sqrt{\frac{\mathcal{L}_\lambda(\theta_{t_0,\lambda})}{\beta}}$ lies in the neighborhood of $\theta_\infty$ where the loss is $\beta$-PL. We can apply Lemma A.2 to obtain that at time $T_\lambda$ the loss will be below $\lambda k_0$ and one can easily check that $k_0$ is bounded as $\lambda \searrow 0$ (since $\theta_{t_0,\lambda}$ converges to the unregularized GF $\theta_{t_0,\lambda=0}$ as $\lambda \searrow 0$). Since the loss will only decrease after that, we know that GF will converge to a local minimum with $O(\lambda)$ loss.

Let us now assume by contradiction that there is a sequence $\lambda_1 > \lambda_2 > \ldots$ with $\lambda_n \to 0$ as $n \to \infty$ such that the minimum $\theta_{\infty,\lambda_n}$ that GF with ridge $\lambda_n$ converges to is rank-underestimating for all $n$, i.e $\text{Rank}A_{\theta_{\infty,\lambda_n}} < \tilde{r}^*$. Since $\lambda_n \|\theta_{\infty,\lambda_n}\| \leq \mathcal{L}_\lambda(\theta_{\infty,\lambda_n}) = O(\lambda_n)$, we know that the $\theta_{\infty,\lambda_n}$ are bounded, which implies the existence of a convergent subsequence that converges to parameters $\tilde{\theta}$ which by continuity of $\theta \mapsto A_\theta$ and $\theta, \lambda \mapsto \mathcal{L}_\lambda(\theta)$ satisfies $\text{Rank}A_{\tilde{\theta}} < \tilde{r}^*$ and $\mathcal{L}(\tilde{\theta}) = 0$, which is in contradiction with the assumption that $r^*$ is the smallest fitting rank. $\qquad\square$

**Lemma A.2.** *Let the loss $\mathcal{L}$ satisfy the $\beta$-PL inequality ($\frac{1}{2} \|\nabla\mathcal{L}(\theta)\|^2 \geq \beta\mathcal{L}(\theta)$) in a ball of radius $R = \frac{\sqrt{\mathcal{L}_\lambda(\theta_0)}}{\sqrt{\beta}}$ around initialization $\theta_0$ for some $\lambda$, then there is a time $T_\lambda \leq -\frac{\log\lambda + \log k_0}{\beta}$ for $k_0 = \frac{2}{\beta}\left(2\sqrt{2\beta\mathcal{L}(\theta_0)}\,(\|\theta_0\| + R) + \beta\,(\|\theta_0\| + R)^2\right)$, where GF $\theta_{T_\lambda,\lambda}$ on the $L_2$-regularized loss $\mathcal{L}_\lambda$ satisfies $\mathcal{L}_\lambda(\theta_{T_\lambda,\lambda}) = \lambda k_0$.*

*Proof.* Let $T_{R,\lambda}$ be the first time gradient flow $\theta_{t,\lambda}$ leaves the ball of radius $R$, we will describe the dynamics before $T_{R,\lambda}$ and then show that $T_{R,\lambda}$ is larger than the time $T_\lambda$ we are interested in.

Inside the ball, we have

$$
\begin{aligned}
\|\nabla\mathcal{L}_\lambda(\theta)\|^2 &\geq (\|\nabla\mathcal{L}(\theta)\| - 2\lambda\|\theta\|)^2 \\
&\geq \left(\sqrt{2\beta\mathcal{L}(\theta)} - 2\lambda\|\theta\|\right)^2 \\
&\geq 2\beta\mathcal{L}(\theta) - 2\sqrt{2\beta\mathcal{L}(\theta)}2\lambda\|\theta\| \\
&\geq 2\beta\mathcal{L}_\lambda(\theta) - 2\lambda\left(2\sqrt{2\beta\mathcal{L}(\theta_0)}\,(\|\theta_0\| + R) + \beta\,(\|\theta_0\| + R)^2\right) \\
&\geq \beta\,(2\mathcal{L}_\lambda(\theta) - \lambda k_0)\,,
\end{aligned}
$$

for $k_0 = \frac{2}{\beta}\left(2\sqrt{2\beta\mathcal{L}(\theta_0)}\,(\|\theta_0\| + R) + \beta\,(\|\theta_0\| + R)^2\right)$.

Let $T_\lambda$ be the first time that $\mathcal{L}_\lambda(\theta_{t,\lambda}) = \lambda k_0$, , then for all $t \leq \min\{T_\lambda, T_{R,\lambda}\}$

$$
\partial_t\mathcal{L}_\lambda(\theta_{t,\lambda}) = -\|\nabla\mathcal{L}_\lambda(\theta_{t,\lambda})\|^2
$$

$$\leq -2\beta \left( \mathcal{L}_\lambda(\theta_{t,\lambda}) - \frac{\lambda k_0}{2} \right)$$

$$\leq -\beta \mathcal{L}_\lambda(\theta_{t,\lambda}),$$

which implies that $\mathcal{L}_\lambda(\theta_{t,\lambda}) \leq \mathcal{L}_\lambda(\theta_0)e^{-\beta t}$ and thus that $T_\lambda \leq -\frac{\log \lambda}{\beta} - \frac{\log k_0/2}{\beta}$ under the condition that this is smaller than $T_{R,\lambda}$.

Let us now show that $T_{R,\lambda} \geq T_\lambda$, by showing that $\|\theta_{T_\lambda,\lambda} - \theta_0\| < R$.

$$\|\theta_{T_\lambda,\lambda} - \theta_0\| \leq \int_0^{T_\lambda} \|\nabla \mathcal{L}_\lambda(\theta_{t,\lambda})\| \, dt$$

$$= \int_0^{\mathcal{L}_\lambda(\theta_0)-\lambda\frac{k_0}{2}} \left\|\nabla \mathcal{L}_\lambda(\theta_{t(\tau),\lambda})\right\|^{-1} d\tau$$

where we did a change of variable in time to $t(\tau)$ which is chosen so that $\mathcal{L}_\lambda(\theta_{t(\tau),\lambda}) = \mathcal{L}_\lambda(\theta_0) - \tau$ which implies that $\partial_\tau t(\tau) = \left\|\nabla \mathcal{L}_\lambda(\theta_{t(\tau),\lambda})\right\|^{-2}$ (so that $\partial_\tau \mathcal{L}_\lambda(\theta_{t(\tau),\lambda}) = -\left\|\nabla \mathcal{L}_\lambda(\theta_{t(\tau),\lambda})\right\|^{2-2} = -1$ as needed). We can now further bound

$$\int_0^{\mathcal{L}_\lambda(\theta_0)-\lambda\frac{k_0}{2}} \left\|\nabla \mathcal{L}_\lambda(\theta_{t(\tau),\lambda})\right\|^{-1} d\tau \leq \int_0^{\mathcal{L}_\lambda(\theta_0)-\lambda\frac{k_0}{2}} \frac{1}{\sqrt{\beta \mathcal{L}_\lambda(\theta_{t,\lambda})}} d\tau$$

$$= \frac{1}{\sqrt{\beta}} \int_0^{\mathcal{L}_\lambda(\theta_0)-\lambda\frac{k_0}{2}} \frac{1}{\sqrt{\mathcal{L}_\lambda(\theta_0)-\tau}} d\tau$$

$$= \frac{1}{\sqrt{\beta}} \left( \sqrt{\mathcal{L}_\lambda(\theta_0)} - \sqrt{\lambda\frac{k_0}{2}} \right).$$

$$\leq \frac{\sqrt{\mathcal{L}_\lambda(\theta_0)}}{\sqrt{\beta}}$$

$$= R.$$

$\square$

While the PL inequality condition might be unexpected, it is actually satisfied at almost all global minima:

**Proposition A.7.** *Given global minimum $\theta$ of a network with widths $w_\ell \geq d_{in} + d_{out}$ for all $\ell = 1, \ldots, L-1$, then for all $\epsilon > 0$ there is a closeby global minimum $\theta'$, i.e. $\|\theta - \theta'\| \leq \epsilon$, such that the loss satisfies the PL inequality in a neighborhood of $\theta'$.*

*Proof.* W.l.o.g., let us assume that $d_{in} \leq d_{out}$, then it is possible to change the parameters infinitesimally to make $W_{L-1} \cdots W_1$ full rank while keeping the outputs $A_\theta = W_L \cdots W_1$ unchanged (by only changing $W_{L-1} \cdots W_1$ orthogonally to $\mathrm{Im} W_L^T$ which is possible since $w_{L-1} \geq d_{in} + d_{out}$).

We now choose a neighborhood of $\theta'$ such that the smallest singular value of $W_{L-1} \cdots W_1$ is lower bounded by some $\lambda > 0$. For any parameters $\theta$ in this neighborhood, the loss satisfies the $\beta = \frac{2\lambda^2}{N}$-PL inequality:

$$\|\nabla \mathcal{L}(\theta)\|^2 = \frac{1}{N^2} \sum_\ell \left\| W_{\ell+1}^T \cdots W_L^T [M \odot (A^* - A_\theta)] W_1^T \cdots W_{\ell-1}^T \right\|^2$$

$$\geq \frac{1}{N^2} \left\| [M \odot (A^* - A_\theta)] W_1^T \cdots W_{L-1}^T \right\|^2$$

$$\geq \frac{\lambda^2}{N^2} \|M \odot (A^* - A_\theta)\|^2$$

$$= \frac{2\lambda^2}{N} \mathcal{L}(\theta).$$

$\square$

The PL-inequality is typically satisfied in the NTK regime (13; 24), but in the Saddle-to-Saddle regime (23; 14) it seems that GF converges to the vicinity of a minima that does not satisfy the PL inequality (minima that are balanced and low-rank typically do not satisfy it), so that the PL-inequality might only be satisfied in a small neighborhood and with a small constant $\beta$. This suggests that in settings where the two notions of minimal rank $r^*$ and $\tilde{r}^*$ do not agree, the question of which minima GF converges to might be dependent on the regime of training we are in, with the NTK regime leading to a rank no less than $\tilde{r}^*$ and the Saddle-to-Saddle regime leading to a rank no less than $r^*$ at least for reasonable values of $\lambda$.

## B  LOW RANK BIAS

In Theorem 5, there are two statements: (1) if $\theta_t \in B_{r,\varepsilon_1,\varepsilon_2,C}$ then $\theta_{t+1} \in B_{r,\varepsilon_1,\varepsilon_2,C}$ and (2) with a positive probability such that there exists a time $T$ such that $\theta_T \in B_{r,\varepsilon_1,\varepsilon_2,C}$. The following theorems give the formal expression of the two statements.

**Theorem B.1.** *For weight $W_l$, $l = 1,\ldots,L$, $L \geq 3$, let*

$$B_{C,\varepsilon_1} := \{\theta : \|W_l\|_F^2 \leq C, \ \|W_l W_l^\top - W_{l+1}^\top W_{l+1}\|_2 \leq \varepsilon_1, \ l = 1,\ldots,L\},$$

*where $C \geq C_1/2\lambda$. Define $F_\alpha(x) = \sum_{i=1}^d f_\alpha(x_i)$ for any $x \in \mathbb{R}^d$, where $f_\alpha$ is twice differentiable function such that $f_\alpha(0) = 0$, $f_\alpha(x) = 1$ for $x > \alpha$, $0 \leq f'_\alpha(x) \leq \frac{K}{\alpha}$ for some constant $K$ and $f''_\alpha(x) \leq 0$. Denote*

$$B_{r,\varepsilon_2} := \{\theta : F_\alpha \circ \sigma(W_l^\top W_l) \leq r + \varepsilon_2, \ l = 1,\ldots,L\},$$

*where $\sigma$ maps a matrix to its singular values and $\alpha \leq \left(\frac{\lambda^2}{2(C_1+C^L)}\right)^{\frac{1}{L-2}}$. Then for any $\varepsilon_1,\varepsilon_2 > 0$ such that $\varepsilon_2 < 1/2$ and $\sqrt{\varepsilon_1} \leq \frac{\lambda\alpha\varepsilon_2}{64nKL(r+1)^2C^{\frac{L-1}{2}}\sqrt{2(C_1+C^L)}}$, if $\theta(t) \in B := B_{C,\varepsilon_1} \cap B_{r,\varepsilon_2}$, then stochastic gradient descent iteration with learning rate*

$$\eta \leq \min\left\{\frac{C_1}{4(2(C_1+C^L)C^{L-1}+\lambda^2 C)}, \frac{2\lambda\varepsilon_1}{4(C_1+C^L)C^{L-1}+\lambda^2\varepsilon_1}, \right.$$
$$\left. \frac{\lambda\alpha\varepsilon_2}{64nK(r+1)^2(2(C_1+C^L)C^{L-1}+\lambda^2 C)}, \frac{2(r+1)}{\lambda}\right\}$$

*satisfies $\theta(t+1) \in B$, where $n$ is the maximal widths and heights of weight matrices.*

**Theorem B.2.** *For any initialization $\theta_0$, denote $C_0 := \max_{1\leq l\leq L}\|W_l\|_F^2$, if*

$$\eta \leq \min\left\{\frac{C_1}{4(2(C_1+C_0^L)C_0^{L-1}+\lambda^2 C_0)}, \frac{\lambda\varepsilon_1}{4(C_1+C^L)C^{L-1}+2\lambda^2 C}\right\}$$

*and $C \geq \frac{C_1}{\lambda}$, then for any time $T = T_0 + T_1$ satisfying $T_0 \geq \frac{\log(2C_0/\varepsilon_1)}{\eta\lambda}$ and $T_1 \geq \frac{\log((4(n-r)C)/(\alpha\varepsilon_2))}{2\eta\lambda}$, we have*

$$\mathbb{P}(\theta_T \in B_{r,\varepsilon_1,\varepsilon_2,C}) \geq \left(\frac{r}{\min\{d_{in}, d_{out}\}}\right)^{T_1}.$$

## C  PRELIMINARIES OF PROOFS

### C.1  FACTS IN LINEAR ALGEBRA

**Fact C.1.** $\|AB\|_* \leq \|A\|_* \|B\|_*$, *where $\|\cdot\|_*$ represents Frobenius norm or 2 norm.*

**Fact C.2.** *For matrices $A, B$ satisfy $AB$ is square, we have $|\mathrm{Tr}(AB)| \leq \|A\|_F \|B\|_F$.*

Let $\sigma_1 \geq \sigma_2 \geq \cdots \geq \sigma_r$ be the singular values of a matrix $A \in \mathbb{R}^{m\times n}$, where $r = \min\{m, n\}$. We have following facts.

**Fact C.3.** $\sigma_i(AB) \leq \sigma_1(A)\sigma_i(B)$ *and* $\sigma_i(AB) \leq \sigma_i(A)\sigma_1(B)$ *for any $i$.*

**Fact C.4** (Theorem 3.3.13 in (31)). *For a square matrix $A \in \mathbb{R}^{n \times n}$, let $\lambda_1 \geq \cdots \geq \lambda_n$ be the eigenvalues. Then we have*

$$\sum_{i=1}^{k} |\lambda_i^p(A)| \leq \sum_{i=1}^{k} \sigma_i^p(A)$$

*for $k = 1, 2, \ldots, n$ and $p > 0$.*

### C.2 SPECTRAL FUNCTION

For a function $f : \mathbb{R}^n \mapsto \mathbb{R}$ that preserves permutation, we consider the function $f \circ \lambda$, where $\lambda(A)$ represents all eigenvalues of symmetric matrix $A \in \mathbb{R}^{n \times n}$. We define $\text{Diag}\mu$ be the diagonal matrix with its entries equal to $\mu$ and $\text{diag}A = (A_{11}, \ldots, A_{nn})$. The following lemmas gives the first and second order derivatives of $f \circ \lambda$.

**Lemma C.1** (Lemma 3.1 from (21)). *$f$ is differentiable at point $\lambda(A)$ if and only if $f \circ \lambda$ is differentiable at $A$. Moreover, we have*

$$\nabla(f \circ \lambda)(A) = U(\text{Diag}\nabla f(\lambda(A)))U^\top,$$

*where $U$ is a orthogonal matrix satisfying $A = U(\text{Diag}\lambda(A)U^\top$.*

For a decreasing sequence $\mu \in \mathbb{R}^n$, where

$$\mu_1 = \cdots = \mu_{k_1} > \mu_{k_1+1} = \cdots = \mu_{k_2} > \mu_{k_2+1} \cdots \mu_{k_r},$$

denote $I_l = \{k_{l-1} + 1, \ldots, k_l\}$ for $l = 1, \ldots, r$. For a twice differentiable function $f$, we define vector $b(\mu)$ as

$$b_i(\mu) = \begin{cases} f_{ii}''(\mu) & \text{if } |I_l| = 1, \\ f_{pp}''(\mu) - f_{pq}''(\mu) & \text{for any } p \neq q \in I_l \end{cases} \tag{1}$$

and matrix $\mathcal{A}(\mu)$ as

$$\mathcal{A}_{ij}(\mu) = \begin{cases} 0 & \text{if } i = j, \\ b_i(\mu) & \text{if } i \neq j \text{ but } i, j \in I_l, \\ \frac{f_i'(\mu) - f_j'(\mu)}{\mu_i - \mu_j} & \text{otherwise.} \end{cases}$$

**Lemma C.2** (Theorem 3.3 from (21)). *$f$ is twice differentiable at point $\lambda(A)$ if and only if $f \circ \lambda$ is twice differentiable at $A$. Moreover, we have*

$$\nabla^2(f \circ \lambda)(A)[H] = \nabla^2 f(\lambda(A))[\text{diag}\tilde{H}, \text{diag}\tilde{H}] + \langle \mathcal{A}(\lambda(A), \tilde{H} \circ \tilde{H}) \rangle,$$

*where $A = W\text{Diag}\lambda(A)W^\top$ and $\tilde{H} = W^\top H W$.*

## D PROOF OF THEOREM B.1

For stochastic gradient descent, the parameter updates as

$$\theta_{t+1} = (1 - \eta\lambda)\theta_t - \frac{\eta}{2}\nabla_\theta(A_{i_t j_t}^* - A_{\theta_t, i_t j_t})^2. \tag{2}$$

Then for each $l$, $l$-th layer's weight $W_l$ updates as

$$W_l(t+1) = W_l(t) - \eta\left(W_{l+1}(t)^\top \cdots W_L(t)^\top G_{\theta_t, i_t j_t} W_1(t)^\top \cdots W_{l-1}(t)^\top + \lambda W_l(t)\right), \tag{3}$$

where $G_{\theta, ij}$ is a matrix where the $(i, j)$-th entry is $A_{\theta, ij} - A_{ij}^*$ and other entries are 0. In the proofs below, we will omit the iteration $t$ for convenience (for example $W_l = W_l(t)$). We denote $T_l = W_{l+1}^\top \cdots W_L^\top G_{\theta, ij} W_1^\top \cdots W_{l-1}^\top$.

### D.1 APPROXIMATE BALANCE

First we give a lemma that bounds $\|G_{\theta,ij}\|_F$.

**Lemma D.1.** *For any $C$, if $\|W_l(t)\|_F^2 \leq C$ for any $l = 1, \ldots, L$, then $\|G_{\theta,ij}\|_F^2 \leq 2(C_1 + C^L)$.*

*Proof.* By Fact C.1, we have $\|A_\theta\|_F^2 \leq \prod_{l=1}^{L} \|W_l\|_F^2 \leq C^L$. Then

$$\begin{aligned}
\|G_{\theta,ij}\|_F^2 = (A_{\theta,ij} - A_{ij}^*)^2 &\leq 2\left(A_{\theta,ij}^2 + (A_{ij}^*)^2\right) \\
&\leq 2\left(\|A_\theta\|_F^2 + C_1\right) \leq 2(C^L + C_1).
\end{aligned} \tag{4}$$

$\square$

**Proposition D.2.** *For any $C \geq \frac{C_1}{2\lambda}$, if $\|W_l(t)\|_F^2 \leq C$ for any $l = 1, \ldots, L$, then stochastic gradient descent iteration with learning rate $\eta \leq \frac{C_1}{4(2(C_1+C^L)C^L+\lambda^2 C)}$ satisfies $\|W_l(t+1)\|_F^2 \leq C$ for $l = 1, \ldots, L$.*

*Proof.* If $\|W_l\|_F^2 \leq C$, we have

$$\|W_l(t+1)\|_F^2 = \|W_l\|_F^2 - 2\eta\left(\lambda\|W_l\|_F^2 + \text{Tr}(W_l^\top T_l)\right) + \eta^2\|T_l + \lambda W_l\|_F^2. \tag{5}$$

We estimate each part in the equation separately. We have

$$\begin{aligned}
\text{Tr}(W_l^\top T_l) &= \text{Tr}(W_1^\top \cdots W_L^\top G_{\theta,ij}) \\
&= \text{Tr}(A_\theta^\top G_{\theta,ij}) = A_{\theta,ij}(A_{\theta,ij} - A_{ij}^*) \\
&\geq -\frac{1}{4}(A_{ij}^*)^2 \geq -\frac{1}{4}C_1.
\end{aligned}$$

By Lemma D.1, we have

$$\begin{aligned}
\|T_l + \lambda W_l\|_F^2 &\leq 2\left(\|G_{\theta,ij}\|_F^2 \prod_{k \neq l} \|W_k\|_F^2 + \lambda^2\|W_l\|_F^2\right) \\
&\leq 2\left(2(C_1 + C^L)C^{L-1} + \lambda^2 C\right).
\end{aligned}$$

Then we have

$$\|W_l(t+1)\|_F^2 \leq (1 - 2\eta\lambda)\|W_l\|_F^2 + \frac{1}{2}\eta C_1 + 2\eta^2(2(C_1 + C^L)C^{L-1} + \lambda^2 C).$$

When $\eta \leq \frac{C_1}{4(2(C_1+C^L)C^{L-1}+\lambda^2 C)}$ and $C \geq \frac{C_1}{2\lambda}$,

$$\|W_l(t+1)\|_F^2 \leq (1 - 2\eta\lambda)\|W_l\|_F^2 + \eta C_1 \leq C.$$

$\square$

**Proposition D.3.** *For any $\varepsilon, C > 0$, if $\|W_l(t)W_l(t)^\top - W_{l+1}(t)^\top W_{l+1}(t)\|_2 \leq \varepsilon$ and $\|W_l(t)\|_F^2 \leq C$ for all $l$, then stochastic gradient descent iteration with learning rate $\eta \leq \frac{2\lambda\varepsilon_1}{4(C_1+C^L)C^{L-1}+\lambda^2\varepsilon_1}$ satisfies $\|W_l(t+1)W_l(t+1)^\top - W_{l+1}(t+1)^\top W_{l+1}(t+1)\|_2 \leq \epsilon$ for $l = 1, \ldots, L$.*

*Proof.* We first compute the update of $W_l W_l^\top$:

$$\begin{aligned}
W_l(t+1)W_l(t+1)^\top &= ((1-\eta\lambda)W_l - \eta T_l)((1-\eta\lambda)W_l - \eta T_l)^\top \\
&= (1-\eta\lambda)^2 W_l W_l^\top - (1-\eta\lambda)\eta(W_l T_l^\top + T_l W_l^\top) + \eta^2 T_l T_l^\top.
\end{aligned}$$

Similarly, we have

$$\begin{aligned}
W_{l+1}&(t+1)^\top W_{l+1}(t+1) \\
&= (1-\eta\lambda)^2 W_{l+1}^\top W_{l+1} - (1-\eta\lambda)\eta(W_{l+1}\top T_{l+1} + T_{l+1}^\top W_{l+1}) + \eta^2 T_{l+1}^\top T_{l+1}.
\end{aligned}$$

Since

$$T_l W_l^\top = W_{l+1}^\top \cdots W_L^\top G_{\theta,ij} W_1^\top \cdots W_l^\top = W_{l+1}^\top T_{l+1}$$

and

$$W_l T_l^\top = \left(T_l W_l^\top\right)^\top = \left(W_{l+1}^\top T_{l+1}\right)^\top = T_{l+1}^\top W_{l+1},$$

we have

$$W_l(t+1)W_l(t+1)^\top - W_{l+1}(t+1)^\top W_{l+1}(t+1)$$
$$= (1-\eta\lambda)^2(W_l W_l^\top - W_{l+1}^\top W_{l+1}) + \eta^2(T_l T_l^\top - T_{l+1}^\top T_{l+1}).$$

Then

$$\|W_l(t+1)W_l(t+1)^\top - W_{l+1}(t+1)^\top W_{l+1}(t+1)\|_2$$
$$\leq (1-\eta\lambda)^2 \left\|W_l W_l^\top - W_{l+1}^\top W_{l+1}\right\|_2 + \eta^2 \left\|T_l T_l^\top - T_{l+1}^\top T_{l+1}\right\|_2$$
$$\leq (1-\eta\lambda)^2\varepsilon + \eta^2\|G_{\theta,ij}\|_2^2 \left(\prod_{k\neq l}\|W_k\|_2^2 + \prod_{k\neq l+1}\|W_k\|_2^2\right) \tag{6}$$
$$\leq \varepsilon - 2\eta\lambda\varepsilon + \eta^2\lambda^2\varepsilon + 4\eta^2(C_1 + C^l)C^{L-1}.$$

When $\eta \leq \frac{2\lambda\varepsilon_1}{4(C_1+C^L)C^{L-1}+\lambda^2\varepsilon_1}$,

$$\|W_l(t+1)W_l(t+1)^\top - W_{l+1}(t+1)^\top W_{l+1}(t+1)\|_2 \leq \varepsilon - 2\eta\lambda\varepsilon + 2\eta\lambda\varepsilon = \varepsilon.$$

$\square$

With Proposition D.2 and Proposition D.3, we have the following statement that $\theta_{t+1}$ is approximate balance and the weight of each layer is bounded.:

**Theorem D.4.** *For any $\varepsilon > 0$ and $C \leq 1/2\lambda$, if $\theta(t) \in B_{C,\varepsilon}$, then the stochastic gradient descent iteration with learning rate $\eta \leq \min\left\{\frac{C_1}{4(2(C_1+C^L)C^{L-1}+\lambda^2 C)}, \frac{2\lambda\varepsilon_1}{4(C_1+C^L)C^{L-1}+\lambda^2\varepsilon_1}\right\}$ satisfies $\theta(t+1) \in B_{C,\varepsilon}$.*

### D.2 APPROXIMATE RANK-$r$

In this section, we prove the following theorem that the weight $W_l(t+1)$ of each layer is approximately rank-$r$.

**Theorem D.5.** *For any $\varepsilon_1, \varepsilon_2 > 0$ such that $\varepsilon_2 < 1/2$ and $\sqrt{\varepsilon_1} \leq \frac{\lambda\alpha\varepsilon_2}{32nL(r+1)C^{\frac{L-1}{2}}\sqrt{2(C_1+C^L)}}$, if the number of layers $L \geq 3$ and $\theta(t) \in B$, then stochastic gradient descent iteration with learning rate $\eta \leq \min\left\{\frac{\lambda\alpha\varepsilon_2}{32n(r+1)(2(C_1+C^L)C^{L-1}+\lambda^2 C)}, \frac{2(r+1)}{\lambda}\right\}$ satisfies $\theta(t+1) \in B_{r,\varepsilon_2}$, where $n$ is the maximal widths and heights of weight matrices.*

*Proof.* We denote by $r_l$ the minima of height and width of $W_l$ and the singular value decomposition $W_l = \tilde{U}_l^\top S_l \tilde{V}_l$, where $\tilde{U}_l$ and $\tilde{V}_l$ are orthogonal matrices. Let $f_\alpha$ and $F_\alpha$ be as defined in Theorem B.1. By Taylor's expansion, for any $l$ we have

$$F_\alpha \circ \sigma\left(W_l(t+1)^\top W_l(t+1)\right)$$
$$= F_\alpha \circ \sigma\left(W_l^\top W_l - \eta W_l^\top T_l - \eta T_l^\top W_l - 2\eta\lambda W_l^\top W_l + \eta^2(T_l + \lambda W_l)^\top(T_l + \lambda W_l)\right)$$
$$= F_\alpha \circ \sigma(W_l^\top W_l) - \left\langle\nabla(F_\alpha \circ \sigma)(W_l^\top W_l), \eta(W_l^\top T_l + T_l^\top W_l)\right\rangle$$
$$\quad - \left\langle\nabla(F_\alpha \circ \sigma(W_l^\top W_l), 2\eta\lambda W_l^\top W_l\right\rangle \tag{7}$$
$$\quad + \left\langle\nabla(F_\alpha \circ \sigma)(W_l^\top W_l), \eta^2(T_l + \lambda W_l)^\top(T_l + \lambda W_l)\right\rangle$$
$$\quad + \nabla^2(F_\alpha \circ \sigma)(W_l^\top W_l + \gamma\eta\Delta)[\eta\Delta, \eta\Delta],$$

where $\gamma \in (0,1)$ and $\Delta = -W_l^\top T_l - T_l^\top W_l - 2\lambda W_l^\top W_l + \eta(T_l + \lambda W_l)^\top(T_l + \lambda W_l)$. By Lemma C.1, we have

$$\nabla(F_\alpha \circ \sigma)(W_l^\top W_l) = \tilde{V}_l^\top \text{diag}\{f_\alpha'(s_1^2), \ldots, f_\alpha'(s_{r_l}^2)\}\tilde{V}_l,$$

where $s_1, \ldots, s_{r_l}$ are the entries of diagonal matrix $S_l$. Denote $\text{diag}\{f_\alpha'(s_1^2), \ldots, f_\alpha'(s_{r_l}^2)\} = f_\alpha'(S^2)$. Then

$$F_\alpha \circ \sigma\left(W_l(t+1)^\top W_l(t+1)\right) \leq \sum_{i=1}^{r_l} f_\alpha(s_i^2) - 2\eta\lambda f_\alpha'(s_i^2)s_i^2 + 2\eta\left|\text{Tr}\left(\tilde{V}_l f_\alpha'(S^2)\tilde{V}_l W_l^\top T_l\right)\right| + \beta,$$

where $\beta$ is the $O(\eta^2)$ term. Now we can estimate the trace term

$$\left| \mathrm{Tr}\left( \tilde{V}_l f'_\alpha(S^2) \tilde{V}_l W_l^\top T_l \right) \right|$$

$$= \left| \mathrm{Tr}\left( W_1^\top \cdots W_{l-1}^\top \tilde{V}_l f'_\alpha(S^2) \tilde{V}_l W_l^\top \cdots W_L^\top G_{\theta,ij} \right) \right|$$

$$\leq \left\| W_1^\top \cdots W_{l-1}^\top \tilde{V}_l f'_\alpha(S^2) \tilde{V}_l W_l^\top \cdots W_L^\top \right\|_F \|G_{\theta,ij}\|_F$$

$$\leq \sqrt{2(C_1 + C^L)} \left\| W_1^\top \cdots W_{l-1}^\top \tilde{V}_l f'_\alpha(S^2) \tilde{V}_l W_l^\top \cdots W_L^\top \right\|_F$$

$$= \sqrt{2(C_1 + C^L)} \sqrt{\mathrm{Tr}\left( (W_1^\top \cdots W_{l-1}^\top \tilde{V}_l f'_\alpha(S^2) \tilde{V}_l W_l^\top \cdots W_L^\top)^\top W_1^\top \cdots W_{l-1}^\top \tilde{V}_l f'_\alpha(S^2) \tilde{V}_l W_l^\top \cdots W_L^\top \right)}$$

$$= \sqrt{2(C_1 + C^L)} \sqrt{\mathrm{Tr}\left( W_{l-1} \cdots W_1 W_1^\top \cdots W_{l-1}^\top \tilde{V}_l f'_\alpha(S^2) \tilde{V}_l W_l^\top \cdots W_L^\top W_L \cdots W_l \tilde{V}_l f'_\alpha(S^2) \tilde{V}_l \right)},$$

$$\tag{8}$$

where the first inequality is from Fact C.2.

Let $E_k = \sum_{i=1}^k (W_{k+1}^\top W_{k+1})^{i-1}(W_k W_k^\top - W_{k+1}^\top W_{k+1})(W_k W_k^\top)^{k-i}$ for $k < l$ and $E_k = \sum_{i=k}^L (W_{k-1} W_{k-1}^\top)^{L-i}(W_k^\top W_k - W_{k+1} W_{k+1}^\top)(W_k^\top W_k)^{i-k}$ for $k > l$. Then we have

$$(W_k W_k^\top)^k = (W_{k+1}^\top W_{k+1})^k + E_k$$

for $k < l$ and

$$(W_k^\top W_k)^{L-k+1} = (W_{k-1} W_{k-1}^\top)^{L-k+1} + E_k$$

for $k > l$. Thus,

$$\mathrm{Tr}\left( W_{l-1} \cdots W_1 W_1^\top \cdots W_{l-1}^\top \tilde{V}_l^\top f'_\alpha(S^2) \tilde{V}_l W_l^\top \cdots W_L^\top W_L \cdots W_l \tilde{V}_l^\top f'_\alpha(S^2) \tilde{V}_l \right)$$

$$\leq \left| \mathrm{Tr}\left( (W_l^\top W_l)^{l-1} \tilde{V}_l^\top f'_\alpha(S^2) \tilde{V}_l (W_l^\top W_l)^{L-l+1} \tilde{V}_l^\top f'_\alpha(S^2) \tilde{V}_l \right) \right| + \sum_{k \neq l} |\mathrm{Tr}(\mathcal{E}_k)| \tag{9}$$

$$= \left| \mathrm{Tr}\left( \tilde{V}_l^\top (f'_\alpha(S^2))^2 S^{2L} \tilde{V}_l \right) \right| + \sum_{k \neq l} |\mathrm{Tr}(\mathcal{E}_k)|,$$

where

$$\mathcal{E}_k = W_{l-1} \cdots W_{k+1} E_k W_{k+1}^\top \cdots W_{l-1}^\top \tilde{V}_l^\top f'_\alpha(S^2) \tilde{V}_l W_l^\top \cdots W_L^\top W_L \cdots W_l \tilde{V}_l^\top f'_\alpha(S^2) \tilde{V}_l$$

for $k < l$ and

$$\mathcal{E}_k = (W_l^\top W_l)^{l-1} \tilde{V}_l^\top f'_\alpha(S^2) \tilde{V}_l W_l^\top \cdots W_{k-1}^\top E_k W_{k-1} W_l \tilde{V}_l^\top f'_\alpha(S^2) \tilde{V}_l$$

for $k > l$.

**Second term in (9).** Denote $\mathcal{E}_k = \mathcal{C}_k \tilde{V}_l^\top f'_\alpha(S^2) \tilde{V}_l$. We define an operator $\mathcal{S}(A)$ equals to the sum of all singular values of $A$. Then by Fact C.4, $|\mathrm{Tr}(\mathcal{E}_k)| \leq \mathcal{S}(\mathcal{E}_k)$. Since $\|W_s\|_2 \leq \|W_s\|_F \leq \sqrt{C}$, $\|E_k\|_2 \leq k\varepsilon_1 C^{k-1}$ for $k < l$ and $\|E_k\|_2 \leq (L-k+1)\varepsilon_1 C^{L-k}$ for $k > l$ by D.4. Since $\|\tilde{V}_l^\top f'_\alpha(S^2)\|_2 \tilde{V}_l \leq \mathrm{Tr}(f'_\alpha(S^2))$, we have $\|\mathcal{C}_k\|_2 \leq \mathrm{Tr}(f'_\alpha(S^2)) C^{L-1} k\varepsilon_1$ for $k < l$ and $\|\mathcal{C}_k\|_2 \leq \mathrm{Tr}(f'_\alpha(S^2)) C^{L-1}(L-k+1)\varepsilon_1$ for $k > l$. Thus, by Fact C.3,

$$\sum_{k \neq l} |\mathrm{Tr}(\mathcal{E}_k)| \leq \mathcal{S}(f'_\alpha(S^2)) \sum_{k \neq l} \|\mathcal{C}_k\|_2$$

$$\leq \mathcal{S}(f'_\alpha(S^2)) \mathrm{Tr}(f'_\alpha(S^2)) C^{L-1} \varepsilon_1 \left( \sum_{k<l} k + \sum_{k>l}(L-k+1) \right) \tag{10}$$

$$\leq C^{L-1} \frac{L^2}{2} \varepsilon_1 \left( \sum_{i=1}^{r_l} f'_\alpha(s_i^2) \right)^2.$$

**First term in** (9). By Fact C.4 and Fact C.3, we have

$$\left| \mathrm{Tr}\left( \tilde{V}_l^\top (f'_\alpha(S^2))^2 S^{2L} \tilde{V}_l \right) \right| \leq \mathcal{S}\left( \tilde{V}_l^\top (f'_\alpha(S^2))^2 S^{2L} \tilde{V}_l \right) \leq \sum_{i=1}^{r_l} (f'_\alpha(s_i^2))^2 s_i^{2L} \leq (\sum_{i=1}^{r_l} f'_\alpha(s_i^2) s_i^L)^2.$$

Then we have that

$$\sqrt{\mathrm{Tr}\left( W_{l-1} \cdots W_1 W_1^\top \cdots W_{l-1}^\top \tilde{V}_l f'_\alpha(S^2) \tilde{V}_l W_l^\top \cdots W_L^\top W_L \cdots W_l \tilde{V}_l f'_\alpha(S^2) \tilde{V}_l \right)}$$

$$\leq \sqrt{(\sum_{i=1}^{r_l} f'_\alpha(s_i^2) s_i^L)^2 + C^{L-1} \frac{L^2}{2} \varepsilon_1 (\sum_{i=1}^{r_l} f'_\alpha(s_i^2))^2}$$

$$\leq \sum_{i=1}^{r_l} f'_\alpha(s_i^2) s_i^L + C^{\frac{L-1}{2}} L \sqrt{\varepsilon_1} \sum_{i=1}^{r_l} f'_\alpha(s_i^2)$$

We denote $\delta := \sum_{i=1}^{r_l} \lambda f'_\alpha(s_i^2) s_i^2 - \sqrt{2(C_1 + C^L)} f'_\alpha(s_i^2) s_i^L$ and $\mathcal{E} := \sqrt{2(C_1 + C^L)} C^{\frac{L-1}{2}} L \sqrt{\varepsilon_1} \sum_{i=1}^{r_l} f'_\alpha(s_i^2)$. Then

$$F_\alpha \circ \sigma\left( W_l(t+1)^\top W_l(t+1) \right) \leq \sum_{i=1}^n f_\alpha(s_i^2) - 2\eta\delta + 2\eta\mathcal{E} + \beta.$$

When $\sum_{i=1}^{r_l} f_\alpha(s_i^2) > r + \varepsilon_2/2$, we have $-2\eta\delta + 2\eta\mathcal{E} + \beta \leq 0$ by Lemma D.7 and when $\sum_{i=1}^{r_l} f_\alpha(s_i^2) \leq r + \varepsilon_2/2$, we have $-2\eta\delta + 2\eta\mathcal{E} + \beta \leq \varepsilon_2/2$ by Lemma D.8. Then $F_\alpha \circ \sigma\left( W_l(t+1)^\top W_l(t+1) \right) \leq r + \varepsilon_2$. □

### D.3 BOUNDS ON ERROR TERMS

Note that we have $f_\alpha(x) \in [0,1]$, $f'_\alpha(x) \in [0, \frac{2}{\alpha}]$ and $f''_\alpha(x) \leq 0$.

**Lemma D.6.** *With same conditions and notations in Theorem D.5, the $O(\eta^2)$ term*

$$\beta \leq 2\eta^2(2(C_1 + C^L)C^{L-1} + \lambda^2 C) \sum_{i=1}^{r_l} f'_\alpha(s_i^2).$$

*Proof.* As defined in Theorem D.5,

$$\beta = \left\langle \nabla(F_\alpha \circ \sigma)(W_l^\top W_l), \eta^2(T_l + \lambda W_l)^\top (T_l + \lambda W_l) \right\rangle \tag{11}$$
$$+ \nabla^2(F_\alpha \circ \sigma)(W_l^\top W_l + \gamma\eta\Delta)[\eta\Delta, \eta\Delta].$$

We bound the two terms in (11) separately.

**First term in** (11). By Fact C.3 and the proof of Proposition D.2, we have

$$\left\langle \nabla(F_\alpha \circ \sigma)(W_l^\top W_l), \eta^2(T_l + \lambda W_l)^\top (T_l + \lambda W_l) \right\rangle$$
$$\leq \eta^2 \mathrm{Tr}(f'_\alpha(S^2)) \|T_l + \lambda W_l\|_2^2 \tag{12}$$
$$\leq 2\eta^2(2(C_1 + C^L)C^{L-1} + \lambda^2 C) \sum_{i=1}^{r_l} f'_\alpha(s_i^2)$$

**Second term in** (11). By Lemma C.2,

$$\nabla^2(F_\alpha \circ \sigma)(W_l^\top W_l + \gamma\eta\Delta)[\eta\Delta, \eta\Delta]$$
$$= \eta^2 \left[ \nabla^2 F_\alpha(\sigma(W_l^\top W_l + \gamma\eta\Delta))[\mathrm{diag}\tilde{\Delta}, \mathrm{diag}\tilde{\Delta}] + \langle \mathcal{A}(\sigma(W_l^\top W_l + \gamma\eta\Delta)), \tilde{\Delta} \circ \tilde{\Delta} \rangle \right]$$

where $\tilde{\Delta} = \tilde{V}_l \Delta \tilde{V}_l^\top$ and

$$\mathcal{A}_{ij}(\sigma(W_l^\top W_l + \gamma\eta\Delta)) = \begin{cases} f''_\alpha(\tilde{s}_i^2) & \text{if } i \neq j \text{ but } \tilde{s}_i^2 = \tilde{s}_j^2, \\ \frac{f'_\alpha(\tilde{s}_i^2) - f'_\alpha(\tilde{s}_j^2)}{\tilde{s}_i^2 - \tilde{s}_j^2} & \text{if } \tilde{s}_i^2 \neq \tilde{s}_j^2, \\ 0 & \text{otherwise,} \end{cases}$$

where $\tilde{s}_1, \ldots \tilde{s}_{r_l}$ are the eigenvalues of $W_l^\top W_l + \gamma\eta\Delta$. Since $\nabla^2 F_\alpha(\sigma(W_l^\top W_l + \theta\eta\Delta)) = \mathrm{diag}\{f_\alpha''(\tilde{s}_1^2), \ldots, f_\alpha''(\tilde{s}_{r_l}^2)\}$ with all entries non-positive, we have

$$\nabla^2 F_\alpha(\sigma(W_l^\top W_l + \gamma\eta\Delta))[\mathrm{diag}\tilde{\Delta}, \mathrm{diag}\tilde{\Delta}] \leq 0.$$

Moreover, since $f_\alpha(x)$ is concave, all entries of $\mathcal{A}(\sigma(W_l^\top W_l))$ are non-positive. Thus,

$$\langle \mathcal{A}(\sigma(W_l^\top W_l + \gamma\eta\Delta)), \tilde{\Delta} \circ \tilde{\Delta}\rangle \leq 0.$$

Overall, $\beta \leq 2\eta^2(2(C_1 + C^L)C^{L-1} + \lambda^2 C)\sum_{i=1}^{r_l} f_\alpha'(s_i^2)$ $\qquad\square$

**Lemma D.7.** *With same conditions and notations in Theorem D.5, when $r + \varepsilon_2/2 < \sum_{i=1}^{r_l} f_\alpha(s_i^2) \leq r + \varepsilon_2$, we have $-2\eta\delta + 2\eta\mathcal{E} + \beta \leq 0$.*

*Proof.* Define $g(x) = f_\alpha'(x)\left(\lambda x - \sqrt{2(C_1 + C^L)}x^{L/2}\right)$ on $x \geq 0$. Then since $L \geq 3$, for $\alpha \leq \left(\frac{\lambda^2}{2(C_1 + C^L)}\right)^{\frac{1}{L-2}}$, $f_\alpha'(x) = 0$ when $x > \alpha$ and $\lambda x - \sqrt{2(C_1 + C^L)}x^{L/2} \geq 0$ when $x \leq \alpha$. Thus, $g(x) \geq 0$ for any $x \geq 0$. Since $\delta = \sum_{i=1}^{r_l} g(s_i^2)$, we have $\delta \geq 0$.

Note that there are at most $r$ $s_i$'s such that $f_\alpha(s_i^2) \geq \frac{2r+1}{2(r+1)} =: M_r$. Otherwise,

$$\sum_{i=1}^{r_l} f_\alpha(s_i^2) \geq (r+1)\frac{2r+1}{2(r+1)} = r + 1/2 > r + \varepsilon_2.$$

Specifically, when $\sum_{i=1}^{r_l} f_\alpha(s_i^2) \geq r + \varepsilon_2/2$, we have

$$\sum_{i: f_\alpha(s_i^2) < M_r} f_\alpha(s_i^2) \geq \sum_{i=1}^{r_l} f_\alpha(s_i^2) - r \geq \frac{\varepsilon_2}{2}. \tag{13}$$

Since $f_\alpha(x)(\alpha - x) \geq 1 - f_\alpha(x)$ by concavity, we have

$$f_\alpha'(x) \geq \frac{1 - f_\alpha(x)}{\alpha - x} \geq \frac{1 - f_\alpha(x)}{\alpha}. \tag{14}$$

We also have $f_\alpha(x) = \int_0^x f_\alpha'(s)ds \leq \frac{Kx}{\alpha}$ according to $f_\alpha(x) \leq \frac{K}{\alpha}$. Then we have

$$x \geq \frac{f_\alpha(x)}{K}\alpha. \tag{15}$$

Moreover, by $L \geq 3$ and the concavity of $f_\alpha(x)$, we have

$$1 - \left(\frac{x}{\alpha}\right)^{\frac{L-2}{2}} \geq 1 - \sqrt{\frac{x}{\alpha}} \geq 1 - \sqrt{f_\alpha(x)} \geq \frac{1 - f_\alpha(x)}{2}. \tag{16}$$

By equation Eq. (14), (15) and (16), we have

$$g(x) = \lambda f_\alpha'(x)x\left(1 - \left(\frac{x}{\alpha}\right)^{\frac{L-2}{2}}\right) \geq \frac{\lambda}{2K}f_\alpha(x)(1 - f_\alpha(x))^2 \geq \frac{\lambda f_\alpha(x)}{8K(r+1)^2}. \tag{17}$$

Thus, we have

$$\delta = \sum_{i=1}^{r_l} g(s_i^2) \geq \sum_{i: f_\alpha(s_i^2) < M_r} g(sx - i^2) > \sum_{i: f(s_i^2) < M_r} f_\alpha(s_i^2)\frac{\lambda}{8K(r+1)^2} \geq \frac{\lambda\varepsilon_2}{16K(r+1)^2}, \tag{18}$$

where the last inequality is by (13). Note that $\sum_{i=1}^{r_l} f_\alpha'(s_i) \leq \frac{2r_l}{\alpha} \leq \frac{2n}{\alpha}$. Then since $\sqrt{\varepsilon_1} \leq \frac{\lambda\alpha\varepsilon_2}{64nKL(r+1)^2 C^{\frac{L-1}{2}}\sqrt{2(C_1 + C^L)}}$, we have $\mathcal{E} \leq \frac{\lambda\varepsilon_2}{32K(r+1)^2}$ and since $\eta \leq \frac{\lambda\alpha\varepsilon_2}{64nK(r+1)^2(2(C_1 + C^L)C^{L-1} + \lambda^2 C)}$, we have $\beta \leq \eta\frac{\lambda\varepsilon_2}{16K(r+1)^2}$. Thus,

$$-2\eta\delta + 2\eta\mathcal{E} + \beta \leq \eta\left(-\frac{\lambda\varepsilon_2}{8K(r+1)^2} + \frac{\lambda\varepsilon_2}{16K(r+1)^2} + \frac{\lambda\varepsilon_2}{16K(r+1)^2}\right) = 0. \tag{19}$$

$\qquad\square$

**Lemma D.8.** *With same conditions and notations in Theorem D.5, when $\sum_{i=1}^{r_l} f_\alpha(s_i^2) \leq r + \varepsilon_2/2$, we have $-2\eta\delta + 2\eta\mathcal{E} + \beta \leq \varepsilon_2/2$.*

*Proof.* Note that $\delta \geq 0$, $\mathcal{E} \leq \frac{\lambda\varepsilon_2}{32K(r+1)^2}$ and $\beta \leq \eta\frac{\lambda\varepsilon_2}{16K(r+1)^2}$. Since $\eta \leq \frac{4K(r+1)^2}{\lambda}$, we have

$$-2\eta\delta + 2\eta\mathcal{E} + \beta \leq \eta\left(\frac{\lambda\varepsilon_2}{16K(r+1)^2} + \frac{\lambda\varepsilon_2}{16K(r+1)^2}\right) \leq \frac{\varepsilon_2}{2}. \tag{20}$$

$\square$

## E  PROOF OF THEOREM B.2

In Theorem B.2, $T = T_0 + T_1$. The following statements explain the change of $\theta_t$ during first $T_0$ iterations and last $T_1$ iterations respectively.

**Theorem E.1.** *For any initialization $\theta_0$, denote $C_0 := \max_{1 \leq l \leq L} \|W_l\|_F^2$, if $\eta \leq \min\left\{\frac{C_1}{4(2(C_1+C_0^L)C_0^{L-1}+\lambda^2 C_0)}, \frac{\lambda\varepsilon_1}{4(C_1+C^L)C^{L-1}+2\lambda^2 C}\right\}$ and $C \geq \frac{C_1}{\lambda}$, then for any time $T \geq \frac{\log(2C_0/\varepsilon_1)}{\eta\lambda}$ we have*

$$\theta_T \in B_{C,\varepsilon_1}$$

*Proof.* Similar to the proof of D.2, if $\eta \leq \frac{C_1}{4(2(C_1+C_0^L)C_0^{L-1}+\lambda^2 C_0)}$ and $C \geq \frac{C_1}{\lambda}$, we have

$$\|W_l(t+1)\|_F^2 \leq (1-2\eta\lambda)\|W_l(t)\|_F^2 + \eta C_1.$$

If $\|W_l(t)\|_F^2 \geq C$, then $\|W_l(t+1)\|_F^2 \leq (1-\eta\lambda)\|W_L(t)\|_F^2$. Otherwise, $\|W_l(t+1)\|_F^2 \leq C$. Thus, there exists $t \leq T_0$ such that $\|W_l\|_F^2 \leq C$ for any $l$ when $T_0 \geq \frac{\log(C_0/C)}{\eta\lambda} \geq \log\left(\frac{C}{C_0}\right)/\log(1-\eta\lambda)$.

After all weights satisfy $\|W_l\|_F^2 \leq C$, $\|W_l W_l^\top - W_{l+1}^\top W_{l+1}\|_2 \leq 2C$. Similar to the proof of D.3, we have

$$\begin{aligned}
\|W_l(t+1)W_l(t+1)^\top &- W_{l+1}(t+1)^\top W_{l+1}(t+1)\|_2 \\
&\leq (1-\eta\lambda)^2\|W_l W_l^\top - W_{l+1}^\top W_{l+1}\|_2 + 4\eta^2(C_1+C^L)C^{L-1} \\
&\leq (1-2\eta\lambda)\|W_l W_l^\top - W_{l+1}^\top W_{l+1}\|_2 + 2\eta^2\lambda^2 C + 4\eta^2(C_1+C^L)C^{L-1}
\end{aligned} \tag{21}$$

When $\eta \leq \frac{\lambda\varepsilon_1}{4(C_1+C^L)C^{L-1}+2\lambda^2 C}$, we have $\|W_l(t+1)W_l(t+1)^\top - W_{l+1}(t+1)^\top W_{l+1}(t+1)\|_2 \leq (1-\eta\lambda)\max\{\|W_l W_l^\top - W_{l+1}^\top W_{l+1}\|_2, \varepsilon_1\}$ for any $l$. Then for $T_1 \geq \frac{\log(2C/\varepsilon_1)}{\eta\lambda} \geq \log\left(\frac{\varepsilon_1}{2C}\right)/\log(1-\eta\lambda)$, $\theta_{T_0+T_1} \in B_{C,\varepsilon_1}$.

**Theorem E.2.** *For any parameter $\theta_t \in B_{\varepsilon_1,C}$ satisfying $\varepsilon_1 \leq \frac{\alpha\varepsilon_2}{2K(n-r)(L-1)}$, then for any $T \geq \frac{\log((2K(n-r)C)/(\alpha\varepsilon_2))}{2\eta\lambda}$ we have*

$$\mathbb{P}(\theta_{t+T} \in B_{r,\varepsilon_1,\varepsilon_2,C}|\theta_t \in B_{\varepsilon_1,C}) \geq \left(\frac{r}{\min\{d_{in},d_{out}\}}\right)^T.$$

*Proof.* For the true matrix $A^*$, the number of columns is $d_{in}$ and the number of rows is $d_{out}$. Let $n = \min\{d_{in}, d_{out}\}$. Without loss of generality we can assume that $n = d_{in}$, i.e. there are $n$ columns. We consider the $r$ columns with most observed entries and denote the the set of these entries by $J$. Then $|J| \geq \frac{r}{n}|I|$ and for each step $s$, the probability of sampling from $J$ is $\mathbb{P}((i_s, j_s) \in J) \geq \frac{r}{n}$. Then the event that all steps $s$ from $t$ to $t + T - 1$, random entries $(i_s, j_s)$ are all sampled from $J$ has probability at least $\left(\frac{r}{n}\right)^T$. Under this event, we consider the weight of first layer $W_1$. We have

$$W_1(s+1) = (1-\eta\lambda)W_1(s) - \eta W_2(s)^\top \cdots W_L(s)^\top G_{\theta_s, i_s j_s}.$$

Then

$$W_1(T) = (1-\eta\lambda)^T W_1(t) + \sum_{s=1}^{T}(1-\eta\lambda)^{T-s} W_2(t+s)^\top \cdots W_L(t+s)^\top G_{\theta_{t+s}, i_{t+s} j_{t+s}}.$$

Since $(i_{t+s}, j_{t+s}) \in J$, the non-zero entry of $G_{\theta_{t+s}, i_{t+s}j_{t+s}}$ is located on the $r$ columns supporting $J$, for any $s = 1, \ldots, T$. Thus, $W_2(t+s)^\top \cdots W_L(t+s)^\top G_{\theta_{t+s}, i_{t+s}j_{t+s}}$ only has non-zero entries on those $r$ columns. Then the $r+1$'s singular value of $W_1(t+T)$ satisfies $\sigma_i(W_1(t+T)) \leq (1-\eta\lambda)^T \sqrt{C}$ for any $i > r$.

For $l > 1$, we have $\|W_{l-1}W_{l-1}^\top - W_l^\top W_l\|_2 \leq \varepsilon_1$. Then $|\sigma_i(W_l^\top W_l) - \sigma_i(W_{l-1}W_{l-1}^\top)| \leq \varepsilon_1$ for any $i$, i.e. $|\sigma_i(W_l)^2 - \sigma_i(W_{l-1})^2|$. Then for any $l$, we have $\sigma_i(W_l(T))^2 \leq (1-\eta\lambda)^{2T}C + (l-1)\varepsilon_1$. When $\varepsilon_1 \leq \frac{\alpha\varepsilon_2}{2K(n-r)(L-1)}$ and $T \geq \frac{\log((2K(n-r)C)/(\alpha\varepsilon_2))}{2\eta\lambda} \geq \log\left(\frac{\alpha\varepsilon_2}{2K(n-r)C}\right)/2\log(1-\eta\lambda)$, we have $\sigma_i(W_l(T))^2 \leq \frac{\alpha\varepsilon_2}{K(n-r)}$ for any $i > r$. Then for any $i > r$,

$$f_\alpha(\sigma_i(W_l(T)^2)) \leq \frac{K}{\alpha}\sigma_i(W_l(T)^2) \leq \frac{\varepsilon_2}{n-r}$$

since $f_\alpha' \leq K/\alpha$. Thus,

$$F_\alpha \circ \sigma(W_l(T)^\top W_l(T)) \leq r + \sum_{i=r+1}^n f_\alpha(\sigma_i(W_l(T)^2)) \leq r + \varepsilon_2.$$

If $n = d_{out}$, the proof is the same by selecting $r$ rows with most observed entries.

## F   Low Rank Property of $A_\theta$

In Proposition 4, we show that for any minimizer $\hat\theta$ in $B_{r,\varepsilon_1,\varepsilon_2,C}$, it is approximate rank-$r$. In fact, any general parameter $\theta \in B_{r,\varepsilon_1,\varepsilon_2,C}$ is approximate rank-$r$ or less:

**Proposition F.1.** *For any parameter $\theta \in B_{r,\varepsilon_1,\varepsilon_2,C}$, we have*

$$\sum_{i=1}^{\text{Rank } A_\theta} f_\alpha(s_i(A_\theta^\top A_\theta)) \leq r + \varepsilon_2 + \frac{KnL^2}{2\alpha}C^{L-1}\varepsilon_1.$$

*Moreover, if $\varepsilon_1 \leq \frac{2\alpha\varepsilon_2}{KnL^2C^{L-1}}$, we have*

$$\sum_{i=1}^{\text{Rank } A_\theta} f_\alpha(s_i(A_\theta^\top A_\theta)) \leq r + 2\varepsilon_2.$$

*Proof.* Since $(W_kW_k^\top)^k = (W_{k+1}^\top W_{k+1})^k + E_k$ and $E_k = \sum_{i=1}^k (W_{k+1}^\top W_{k+1})^{i-1}(W_kW_k^\top - W_{k+1}^\top W_{k+1})(W_kW_k^\top)^{k-i}$, we have

$$A_\theta A_\theta^\top = W_L \cdots W_1 W_1^\top \cdots W_L^\top = (W_LW_L^\top)^L + \sum_{k=1}^{L-1} \mathcal{E}_k,$$

where $\mathcal{E}_k = W_L \cdots W_{k+1} E_k W_{k+1}^\top \cdots W_L^\top$. Then by Taylor's expansion, we have

$$
\begin{aligned}
\sum_{i=1}^{r_L} f_\alpha(s_i(A_\theta^\top A_\theta)) &= F_\alpha \circ \sigma(A_\theta A_\theta^\top) \\
&= F_\alpha \circ \sigma(((W_LW_L^\top)^L + \mathcal{E})) \\
&= F_\alpha \circ \sigma((W_LW_L^\top)^L) + \langle \nabla(F_\alpha \circ \sigma)((W_LW_L^\top)^L), \mathcal{E} \rangle \\
&\quad + \nabla^2(F_\alpha \circ \sigma)((W_LW_L^\top)^L + \gamma\mathcal{E})[\mathcal{E}, \mathcal{E}],
\end{aligned}
\tag{22}
$$

where $\gamma \in (0,1)$ and $\mathcal{E} = \sum_{k=1}^{L-1} \mathcal{E}_k$. Note that the Rank $A_\theta \leq r_L$, so we can let $s_i(A_\theta^\top A_\theta) = 0$ for $i > \text{Rank } A_\theta$.

**First term in** (22).   $F_\alpha \circ \sigma((W_LW_L^\top)^L) = \sum_{i=1}^{r_L} f_\alpha(s_i^{2L})$, where $\{s_1, \ldots, s_{r_L}\}$ are the singular values of $W_l$. Then since $f_\alpha$ is non-decreasing, $f_\alpha(s_i^{2L}) \leq f_\alpha(s_i^2)$ for $s_i \leq 1$ and $f_\alpha(s_i^{2L}) = f_\alpha(s_i^2) = 1$ for $s_i > 1 > \alpha$. Thus, $F_\alpha \circ \sigma((W_LW_L^\top)^L) \leq F_\alpha \circ \sigma(W_L^\top W_L) \leq r + \varepsilon_2$.

**Second term in** (22).  By Lemma C.1, $\nabla(F_\alpha \circ \sigma)((W_L W_L^\top)^L) = \tilde{U}_L \mathrm{diag}\{f'_\alpha(s_i^{2L}), \ldots, f'_\alpha(s_{r_l}^{2L})\}$. Then by Fact C.3 and C.4, we have

$$\left\langle \nabla(F_\alpha \circ \sigma)((W_L W_L^\top)^L), \mathcal{E} \right\rangle \le \mathcal{S}\left( \nabla(F_\alpha \circ \sigma)((W_L W_L^\top)^L) \right) \|\mathcal{E}\|_2$$

$$\le \|\mathcal{E}\|_2 \sum_{i=1}^{r_L} f'_\alpha(s_i^{2L})$$

$$\le \frac{K r_L}{\alpha} \|\mathcal{E}\|_2 \le \frac{K n}{\alpha} \|\mathcal{E}\|_2$$

Since $\|E_k\|_F \le k C^{k-1} \varepsilon_1$, we have $\|\mathcal{E}_k\|_F \le k C^{L-1} \varepsilon_1$. Then

$$\|\mathcal{E}\|_2 \le \|\mathcal{E}\|_F \le \sum_{k=1}^{L-1} \|\mathcal{E}_k\|_F \le C^{L-1} \varepsilon_1 \sum_{k=1}^{L-1} k \le \frac{L^2}{2} C^{L-1} \varepsilon_1.$$

Thus, $\left\langle \nabla(F_\alpha \circ \sigma)((W_L W_L^\top)^L), \mathcal{E} \right\rangle \le \frac{K n L^2}{2\alpha} C^{L-1} \varepsilon_1$.

**Third term in** (22).  By Lemma C.2,

$$\nabla^2(F_\alpha \circ \sigma)((W_L W_L^\top)^L + \gamma \mathcal{E})[\mathcal{E}, \mathcal{E}]$$
$$= \nabla^2 F_\alpha \left( \sigma((W_L W_L^\top)^L + \gamma \mathcal{E}) \right) [\mathrm{diag}\tilde{\mathcal{E}}, \mathrm{diag}\tilde{\mathcal{E}}] + \left\langle \mathcal{A}(\sigma((W_L W_L^\top)^L + \gamma \mathcal{E})), \tilde{\mathcal{E}} \circ \tilde{\mathcal{E}} \right\rangle,$$

where $\tilde{\mathcal{E}} = \tilde{U}_L \mathcal{E} \tilde{U}_L^\top$. Since the entries of $\nabla^2 F_\alpha \left( \sigma((W_L W_L^\top)^L + \gamma \mathcal{E}) \right)$ and $\mathcal{A}(\sigma((W_L W_L^\top)^L + \gamma \mathcal{E}))$ are all non-positive (follows the proof in D.6), we have

$$\nabla^2(F_\alpha \circ \sigma)((W_L W_L^\top)^L + \gamma \mathcal{E})[\mathcal{E}, \mathcal{E}] \le 0.$$

Therefore, we can add the three terms up and have

$$\sum_{i=1}^{\mathrm{Rank}\, A_\theta} f_\alpha(s_i(A_\theta^\top A_\theta)) \le r + \varepsilon_2 + \frac{K n L^2}{2\alpha} C^{L-1} \varepsilon_1.$$

