# OpenReview forum: "Implicit bias of SGD in $L_2$-regularized linear DNNs: One-way jumps from high to low rank"
_ICLR.cc/2024/Conference — ICLR 2024 spotlight_

### Official Review · Reviewer_Gs3p · 2023-10-30

**Soundness:** 3 good
**Presentation:** 2 fair
**Contribution:** 3 good
**Rating:** 8
**Confidence:** 4

**Summary:**

This papers considers the problem of minimizing parameters of a deep linear network with the matrix completion loss. Specifically the authors considered an $\ell_2$ regularized version of this problem and show that with certain conditions on the learning rate and the regularization parameter, SGD can jump from a high-rank local min to a low-rank local min, while it cannot jump from a low-rank local min to a high-rank local min.

**Strengths:**

I think the theoretical results of this paper are interesting because it gives a nice characterization of the implicit bias of GD/SGD for deep linear networks. The proof that SGD can "jump" from high-rank to low-rank local minima is new to me and I think its a good step towards understand the training dynamics for deep neural networks.

**Weaknesses:**

I think the main weakness of this paper is that both the theoretical results and the experiments require specific conditions on the $\ell_2$ regularization parameter $\lambda$ and the learning rate $\eta$, which seem to be a bit artificial. For example, the requirement that $\lambda$ is large in Theorem 3.2 seem to artificially cause $\|\theta\|$ to decay more quickly, thus biasing it towards low numerical rank. In the numerical experiments, a similar annealing technique is used run SGD with a large $\lambda$ and $\eta$, before switching to smaller parameters.

My main point is that the implicit bias observed in this paper could be a result of a deliberate choice of parameters, instead of a natural property of SGD and GD. I hope the authors can clarify this point.

**Questions:**

Please see previous section. Also, I wonder if the proof in this paper for Theorem 3.2 also works for GD? In other words, is $B_r$ also absorbing for GD?

---

> ### Author Response · Authors · 2023-11-16
>
> Thank you for the thoughtful review.
>
> First note that we actually do not need $\lambda$ to be large for our theorem, only $C$ needs to be large. We apologize for the confusion, we did not realize that our statement was confusing, and we have replaced it with "for $r\geq0$, any $\lambda\geq 0$, any large enough $C$ and small enough $\alpha,\epsilon_1,\epsilon_2,\eta$" which can only be understood in the correct way.
>
> This should resolve your concern that the effect we observe is only a result of a very large regularization, since our result applies to arbitrarily small ridge $\lambda$.
>
> Note also that an upper bound on $\eta$ is unavoidable, since large $\eta$ could lead to divergence. Now it seems likely that the bound we require on $\eta$ is not tight, especially w.r.t. its dependence on $\lambda$, and this could be improved in future work.
>
> From our theorem, one can identify settings (for $\lambda,\eta$ small enough) where GD can get stuck at a rank-overestimating minima, while SGD will be able to escape it given enough time, showing a fundamental difference between the two regimes.
>
> Regarding your question. Yes, the sets $B_r$ are absorbing for GD too, only the jumps are specific to SGD. But both are needed to guarantee a gradual decrease of the rank.

---

> > ### Comment · Reviewer_Gs3p · 2023-11-17
> >
> > I thank the authors for addressing my concerns. I believe that this is a good paper that should be accepted. I have raised my score.

---

### Official Review · Reviewer_fYKu · 2023-10-31

**Soundness:** 3 good
**Presentation:** 3 good
**Contribution:** 4 excellent
**Rating:** 8
**Confidence:** 3

**Summary:**

This paper studies the optimization landscape of regularized stochastic gradient descent applied to matrix completion with linear networks problems (which is equivalent to matrix completion with a $2/L$ Shatten norm regularizer). Several properties of the optimization landscape are proved, including the fact that the only critical points of the optimization problem over the factor matrices (minimizing $\mathcal{L}_{\lambda}(\theta)$ must be local minima of the optimization problem over the full matrix $A$, unless they are strict saddle points in the original problem.

 In addition, it is shown that if gradient flow converges to a global minimum, then a version of gradient flow with a sufficiently small regularization parameter will converge to a minimum with a larger rank than the ground truth. Arguably the most significant final result is that stochastic gradient descent jumps from high rank local minima to lower rank local minima, with the jumps being one directional: one cannot return to a higher rank region after entering a lower rank region. Here, the lower rank regions should be understood as defined on page 5, in an approximate sense. Throughout the proofs, the fact that the local minima of the optimization problem over $\mathcal{L}_\lambda(\theta)$ must be balanced (cf. Proposition A.1). An approximate version of this condition is also present in the definition of the low absorbing low rank spaces in the main results of the paper.

**Strengths:**

The main paper is well-written and the results appear generally sound. The results are of great importance to the field and interest to the community. **This is highly non trivial and important work**.

**Weaknesses:**

Although the main paper is well written, the **proofs are not reader-friendly** at all.

The writing of the proofs is very terse and laconic, omitting many details. Although this is reminiscent of some great pure mathematics papers that were ahead of their time and I enjoyed the challenge some of the time, I strongly believe this style should only be considered acceptable if there is absolutely zero tolerance for any errors or inaccuracies whatsoever. I don't think the proofs actually stand up to this amount of scrutiny: there are **at least a few typos, minor errors and imprecisions** in the subset of the proofs I was able to look at, and since a lot of information is left out for the reader to figure out, the additional presence of even a small number of actual errors dramatically expands the "search space" from the point of view of the reader.  I would really like to see a substantial revision of the paper with more detailed and careful proofs (and maintaining my score is conditional on that).

For instance, in page 12, point "(0)", the definition of the $U_i, V_i$ is not really consistent: the index under the $U,V,S$ is used both to mean the iteration step in the sequence and the position in the product $W^L...W^1$.

In addition, in page 13, consider the following statement the authors make " as $\lambda\rightarrow 0$, the critical points of the loss move continuously. Consider a continuous path of critical points, as $\lambda\rightarrow 0$, it converges to..."
Although the argument makes sense intuitively, filling in the gaps with rigorous proofs is definitely beyond the scope of what can be expected of the reader to do. At least some citations are a minimum. I doubt that simple continuity is enough to guarantee convergence (even if a subsequence converges, the path could oscillate widely), probably the only way to rigorously prove the statement is to use a quantitative version of the statement relying on calculus of variations.

This is not the only example. In point (1) in page 14, the authors say "the singular value ..... must converge to a non zero eigenvalue".  It is not clear **why this is the case**, or why the the *singular value* turns into an *eigenvalue* after convergence. Far more details are required.

In the middle of page 14, it is hard to imagine that the equation $U_{\ell,i}(\lambda)U_{\ell-1,i}(\lambda)$ can be correct without **at least a transpose missing**. Of course, the lack of a rigorous and consistent definition of $U_{\ell,i}$ does not help here.

At the bottom of page 14: the line starting with "other directions" ends with " $L-1,)$" and a few lines below we have the equation $U_\ell^\top dU_{\ell}+ = -dU_\ell^\top U_\ell$. What does "+=" mean here? The same issue is present in many other parts of the paper, including in the third line of text on page 15.

Towards the end of Appendix A in page 17, the term "saddle to saddle" is mentioned with absolutely no explanation or citation.

In the middle of page 13, the authors use the fact that "a matrix cannot be approached with matrices of strictly lower rank", which is true but should probably warrant a citation since the equivalent statement is not true for tensors.


The proof of Proposition A4 is very hard to make sense of without further information: the first sentence is ""let A(\lambda) be path of global minima restricted to the set of matrices of rank $r^*$ or less." how do you construct the path? Even for $L=2$, there can be a continuous set of global minima of local intrinsic dimension higher than 2, how do you use the axiom of choice to construct a "path"?
Sentences such as "going along directions that increase the rank of $A(\lambda)$, the regularization term increases at a rate of $d^{2/L}$ for $d$ the distance" definitely need more mathematical details.

Similarly, the statement about $\phi$ being differentiable in the directions which do not change the rank should be made more precise (although I agree with it, probably at least a citation to [1] is a minimum)

For proposition A.5, the proof starts with the following sentence "We know that L2 regularized GF $\theta_\lambda(t)$ converges to unregularized GF $\theta(t)$ as $\lambda\rightarrow 0$". There are two parameters here, $\lambda$ and $t$, is the convergence uniform over all $t$?













========more minor points:=====

Many apologies if I am being picky but as a relative outsider to optimization literature, even the statement that the point 0 is a critical point was not immediately  obvious to me (perhaps either a calculation of the gradient or a mention of the fact that $L>1$ would help).

In the bottom of page 13, the equation before equation (1) is presumably the end of a sentence, thus the next line should be rewritten. Below, that "no such thing happen" should be "no such thing happens"

Some citation for Fact C.4 (Ky Fan?) would be nice.


In page 19, just before the beginning of Section D.1. Do the authors mean $G_{\theta,ij}$ instead of $G_{\theta,j}$?


Just above equation (6), $\|W_\ell|^2$ should be $\|W_\ell\|^2$ and the sentence is missing a period.








[1] Characterization of the subdifferential of some matrix norms, G.A. Watson. 1992, linear algebra and its applications.

**Questions:**

1. In the third line of page 13, ou mention that the quantity in the limit is strictly positive but possibly infinite. Apologies if I  lack some background knowledge but could you explain your reasoning there? It is not at all obvious to me.

2 At the beginning of the proof of proposition A5 in the first equation, should the infimum run over  $Rank A>r$ instead of $Rank A<r $ as written?

---

> ### Author Response · Authors · 2023-11-16
>
> Thank you for the thorough and detailed review.
>
> Your remarks were really valuable to improve the readability of our proofs. We have made many changes that should address your issues and we are still working on improving readability.
>
> We have replaced $U_i,V_i$ by $V_i,W_i$ to avoid the conflict in notation with the $U_\ell$s.
>
> We have removed the reliance of the proofs on a continuous path of minima as $\lambda \searrow 0$. As you said, some work needs to be done to make such statement rigorous and it turns out to not be necessary.
>
> The "eigenvalue" was a typo, it should have been "singular value", and we have changed this part of the proof in the newer version anyway, since we do not rely on a path of minimizers.
>
> You were right that a transpose was missing and we have added a definition of $U_{\ell,i}$.
>
> The "+=" was a typo, we replaced it with "=".
>
> The saddle-to-saddle regime refers to the two papers cited at the end of the sentence, we have moved the citations to earlier in the sentence to clarify this.
>
> We have added a short explanation for why a matrix cannot be approached by matrices of lower rank (the error will always be at least the $r+1$ singular value of $A$).
>
> We have removed the reliance on a path of minima, instead we just take $A(\lambda)$ to be a global minimum of $C_\lambda$ amongst matrices of rank $r^*$ or less.
>
> Thanks for the reference you provided for the differentiability along directions that do not change the rank, we will add it as a justification.
>
> The convergence is not uniform in $t$, but we only need sequential convergence:  for all fixed $t$, we have convergence as $\lambda \searrow 0$. We have made this clearer.
>
> -----------
>
> We will add a short explanation: the first $L-1$ derivatives of the unregularized loss $\mathcal{L}$ vanish at $\theta=0$, thus adding a regularization terms $\lambda\Vert\theta\Vert^2$ leads to a local minimum.
>
> We fixed the typos and error you mentioned and added a citation for Fact C.4.
>
> Regarding your questions:
> 1. In general you get an explosion if $A_{\theta_i}$ approaches $A_{\hat{\theta}}$ from a higher rank direction. Assume that $A_{\theta_i}$ equals to $A_{\hat{\theta}}$ with a small additional singular value: $A_{\theta_i} = A_{\hat{\theta}} + uv^T / i$, then for balanced parameters $\Vert \theta_i - \hat{\theta} \Vert^2 = i^{-\frac{2}{L}}$, leading to an infinite limit.
>
> 2. No, we do want the infimum over $\mathrm{Rank}A<r^*$. Since we know it will be strictly larger than $0$, we can always choose $t_0$ large enough and $\lambda$ small enough to be below this threshold. Once we are below this threshold, we know that the rank of $A_{\theta(t)}$ cannot be strictly smaller than $r^*$.

---

> > ### Comment · Reviewer_fYKu · 2023-11-22
> > **More polishing**
> >
> > Thanks for your answers. I agree the deadline is a bit tight for such a technical paper. I am trying to make my way through your revision since I might not be able to read everything before the deadline I will leave a quick comment here already.
> >
> > I think the notation in the proof of proposition A.2. is still not ideal: you are still using the same symbol ($W_i$) to mean two things: both the matrix of right singular vector of $A_i$ (**ith iteration**) **and** the weight matrix at the ith **layer**. The first sentence of point (0) is also not a sentence.
> >
> > I agree with reviewer NbUi that you absolutely should not make statements about GD if you prove something for GF.
> >
> > Can I ask if there is actually a **proof of Theorem 3.2** (not a sketch) somewhere in the appendix? What about **Theorems B.1 and B.2**?
> >
> >
> > You might want to sit down and check the notations of the whole paper.  Also can we have a revision with the modified parts in a different color for ease of reference?
> >
> >
> > Like I said before, you should add more details for some of the proofs. It's a nice paper but it's hard to parse. One more example: second to last paragraph of the proof of proposition A.3:  "one can guarantee that the second term dominates the third one since C is differentiable." I agree the statement holds, but it is always nice to add details (especially in a highly technical paper with errors). I think the point is that this statement feels counter intuitive because of the lack of incoherence assumptions: I think the size of the neighborhood in which the statement holds can be small in a way that depends on the size of the matrix, which fortunately is not a problem.
> >
> > Regarding my question 1: thanks for the explanation. I agree with it intuitively and it helps already, but I think it's no substitute for a rigorous proof (not just in the case where we add a rank 1 matrix which is the illustrative case you gave in this answer), which should be both here and in the paper. This question is actually key to the proof of Part 2.a of Proposition A.2 as well

---

> > > ### Author Response · Authors · 2023-11-22
> > >
> > > You are right with the notations of proposition A.2, we have changed them to $\tilde{U}_i,\tilde{V}_i$, everything should be fine now. We also fixed the sentence.
> > >
> > > We have updated the proofs to work with GD with a small enough learning rate.
> > >
> > > In the appendix, Theorem 3.2 is split into two theorems B.1 and B.2, which are each proven in their own sections.
> > >
> > > We have put our last changes in red in the newest version, hope this helps.
> > >
> > > We have added a bit more details in the proof of proposition A.3
> > >
> > > Sorry, we misunderstood your question, we thought you asked under which condition it can be infinite. We have added a more detailed argument using the differentiability.

---

### Official Review · Reviewer_NbUi · 2023-10-31

**Soundness:** 2 fair
**Presentation:** 3 good
**Contribution:** 2 fair
**Rating:** 5
**Confidence:** 4

**Summary:**

This paper analyzes the matrix completion task with deep linear neural networks. It shows that the critical points that are not local minima can be avoided with a small ridge. And it shows that GD cannot avoid overestimating minima but  SGD can jump from any minimum to a lower rank minimum.

**Strengths:**

I think analyzing the training dynamics of the deep linear neural networks on the matrix completion task is a very interesting problem. This paper provides insights on the advantages of using SGD to get a low-rank minima and provides experimental results.

**Weaknesses:**

1. I feel the statement of theorems is not very clear. It uses a lot of ''small enough'', "large enough".  I think the statement should be more rigorous.

2. This paper claims that it shows GD can avoid rank-underestimating minima by taking a small enough ridge $\lambda$. But Proposition 3.2 is for Gradient Flow with a very strong assumption. I believe there is a gap.

3. For the function $f_\alpha$, it takes a very specific form. The authors claim that changing $f_\alpha$ with similar properties should not affect the results. I don't see the reason why the condition cannot be extended to more general functions. I believe it could improve the results.

4. A small suggestion is that the proof in the appendix is not easy to read. I think it can be more organized and add more explanation.

**Questions:**

1. In the remark before section 3.2, it's said that it's possible that GD can recover the minimal rank solution easily. Can you say something about this case?

2. I have a concern about the constant in Theorem 3.2. It is said $\lambda$ and $C$ are large enough, but an example of acceptable rates in terms of $\lambda$ is $C \sim \lambda^{-1}$. Is it contradictory?

3. In the proof of Theorem 3.2, r columns with the most observed entries are taken.  What if all the columns have the same number of observed entries? Will the $d_{out}-r$ other columns of $W_L$ decay exponentially? I don't see the relation between rank $r$ and the number of observed entries here.

---

> ### Author Response · Authors · 2023-11-16
>
> Thank you for the thoughtful review. Regarding your points:
> 1. The statement "for small enough $x$" is simply a (commonly used) shortcut for "there is a $x_0$ such that for all $x\leq x_0$". They are both rigorous formal statements. Note that exact bounds are given in the appendix, we did not put them in the main to improve readability.
>
> 2. Since the loss $\mathcal{L}_\lambda$ is smooth, GD (and SGD) converges to GF as $\eta \searrow 0$. We stated the Proposition for GF to simplify the statement.
> We preferred to assume an initialization that converges to a global minimum directly for the following reason. There exist many papers that can guarantee convergence to a global minimum (e.g. in the NTK regime, or with specific initialization, we will add references to such results in the main after the statement of this Proposition), and since it seems that we have not yet fully answered the question of convergence of DLNs, we expect more result to appear in the next years. Our Proposition can be applied to any such settings.
>
> 3. We feel that our proofs are already quite complex and technical, and considering a general $f_\alpha$ would lead to even more complexity for only a small increase in generality.
>
> 4. We have clarified the proofs thanks to the remarks of the reviewers. Please tell us if there is one section in particular that you find difficult to understand, so that we may focus our efforts there.
>
>
> Regarding your questions:
> 1. We say that there might not be any rank-overestimating minima (in contrast to the fact that there always exist rank-underestimating minima, e.g. the origin $\theta=0$), in which case GD with a small enough ridge $\lambda$ and learning rate $\eta$ will converge to a minimum with the right rank by Proposition 3.2. Of course we do not know under what condition this happens, and it might be difficult to characterize, a trivial example would be if we observe all entries of the matrix.
>
> 2. Sorry, we did not realize that our statement was confusing, we replaced it with "for $r\geq0$, any $\lambda\leq 0$, any large enough $C$ and small enough $\alpha,\epsilon_1,\epsilon_2,\eta$" which can only be understood in the correct way. Namely, we do not need $\lambda$ to be large for our theorem, only $C$ needs to be large.
>
> 3. We can guarantee a jump under the event that SGD randomly selects observations only from the same $r$ rows/columns over a large period of time. This could happen for any set of $r$ rows/columns, but we only consider the $r$ rows/columns with the most observed entries since it is most likely to happen to these rows entries. When all rows have the same amount of observed entries, this event is as likely for all subsets of rows/columns (with probability equal to the lower bound given in the statement of the Theorem).

---

> > ### Comment · Reviewer_NbUi · 2023-11-21
> >
> > Thank the authors for the reply.  I have a few more comments on your response.
> >
> > 1. Thanks for clarifying "small enough" and "large enough".  Personally, I would prefer to have explicit bounds in the statement, or at least in the main text. This improves clarity and avoids misinterpretation.
> >
> > 2.  I  don't think it is proper to make claims on GD while the results are for GF. Though GD converges to GF with the learning rate going to $0$, they are different in principle.
> >
> > 3. I am not sure a general $f_\alpha$ would make the proof much more complicated. I think it is an important quantity however currently it is unclear why such $f_\alpha$ is chosen. If a proof for general $f_\alpha$ is not possible, at least the paper should discuss what kind of $f_\alpha$ is reasonable and how it affects the proof.
> >
> > 4. I read the updated draft but still find many arguments in the proofs are vague and handwavy.  E.g., in the proof for Proposition A.5, it is unclear how to apply Lemma A.1 to get the results given Lemma A.1 is a bit complicated, and what $\lambda_n \rightarrow 0$ means (as $n$ goes to infinity?). There are some more.  I believe the proofs should be correct, but my point is that more explanation and details should be included for readability and reproducibility.

---

> > > ### Author Response · Authors · 2023-11-21
> > >
> > > 1. We will add a bit more details to the paragraph that discusses how $C,\alpha,\epsilon_1,\epsilon_2,\eta$ can be chosen as a function of $\lambda$.
> > > 2. We are working on extending the results to GD with a small enough learning rate, there won't be any issue since the loss is Lipshitz inside any ball and one can guarantee that GD will remain inside one such ball. We hope to be able to upload a new version with the updated statements before the end of the discussion period.
> > > 3. We are also working on generalizing the proofs for a more general choice of $f_\alpha$, but we are not sure it's possible by the end of the discussion period tomorrow, but it's definitely possible in a few weeks for the final version.
> > > 4. We have improved the proof of Proposition A.5, thanks for pointing out these issues, there was a mismatch between how Lemma A.1 was used in the proof of Proposition A.5 and the actual statement of Lemma A.1. The updated proof will be updated soon. We have also specified that we are considering the limit as $n\to\infty$ and added more details in general.

---

### Official Review · Reviewer_M8aA · 2023-11-01

**Soundness:** 4 excellent
**Presentation:** 4 excellent
**Contribution:** 4 excellent
**Rating:** 10
**Confidence:** 3

**Summary:**

This work shows that when applied to matrix completion with deep linear networks, SGD can transition from local minima with higher ranks to solutions with lower ranks, while transitions in the opposite direction are zero. Crucially, this results depends on the gradient distribution of SGD which leads to drastically different outcomes than what common SDE-based models for SGD exhibit. The authors further provide numerical experiments that exhibit the predicted transitions in practice.

**Strengths:**

This work provides an interesting theoretical insight on the distinction between stochastic and deterministic gradient descent. I find it especially exciting that it provides a concrete example how the gradient distribution of SGD enables phenomena that are not apparent from SDE-based models.

**Weaknesses:**

I can not think of a weakness. It's a very nice paper in my opinion.

**Questions:**

As you point out, the common Langevin-based models predict a transition among each pair of points and thus fall short to capture the phenomenon shown by your result. Do you know if they would nevertheless exhibit a systematic low-rank bias (i.e. the transition toward lower-rank solutions being much more likely than toward higher-rank solutions)?

---

> ### Author Response · Authors · 2023-11-16
>
> Thanks for the review!
>
> Regarding your question, we would also observe a low-rank bias with Langevin dynamics since minima with a lower rank generally have a smaller parameter norm and thus a smaller regularized loss. It should therefore be more likely to jump from a high loss to a low loss minima than the other way round. Now the relation between parameter norm and rank is not exact and there can be example where the minimal parameter norm solution has a higher rank than another solution with a larger parameter norm (this happens for $\epsilon$ small enough in the setup of our numerical experiments). So the low-rank bias would be in some sense weaker.
>
> But Langevin dynamics do not capture the covariance of the noise of SGD. One can instead study an SDE with matching covariance. But this covariance is not full rank especially inside and around the $B_r$ sets, our intuition is that there is a lot noise along direction that keep or lower the rank, but (almost) no noise along direction that increase the rank. It is still unclear whether a similarcould be proven in this setting (e. .whether the limiting distribution is also supported inside $B_1$).

---

### Author Response · Authors · 2023-11-16
**Correcting an error**

While working on cleaning up the proofs, we have realized that one of the minor result was actually incorrect. Namely, the statement in Theorem 3.1 that for small enough ridge, the loss landscape has only local minima and strict saddles. We believe that a loss could be constructed that would yield a non-strict saddle even for very small $\lambda$. But this error does not affect any other statement and only changes a little bit the overall picture.

The possible existence of non-strict saddles means that we need to ensure that they are avoided. The argument that we have given for avoiding rank-underestimating local minima, by taking a small enough ridge $\lambda$, also works for avoiding any rank-underestimating critical point. And similarly the jumps that we describe also leads SGD to avoid any rank-overestimating critical point. Thus the only thing that we need to change is to switch from writing 'rank-overestimating minima' to 'rank-overestimating critical point' and other similar changes.

Thus the only non-strict saddles that could be hard to avoid are those that have the right rank $r^*$, but are not optimal amongst same rank matrices. But note that there could also be multiple local minima with the right rank $r^*$, some of whom might not fit the training data, hence this problem was not solved in the previous version either. Since we mostly focus on recovering the right rank, we did not think to mention this possible issue, but we decided to add a paragraph discussing it for completeness.

---

### Meta-Review · Area_Chair_oeZm · 2023-12-07

**Metareview:**

This paper studies the performance of SGD on deep linear networks with L2-regularized loss. shows that Stochastic Gradient Descent (SGD) offers a probability of transitioning from a higher-rank minimum to a lower-rank one, but no chance of reverting. It defines sets of minima for SGD, ensuring convergence to minima with equal or lower rank under specific conditions.


The paper underwent review by four experts, and their collective assessment was positive. I concur with the reviewers and share their view that the paper presents a novel and valuable approach to an important problem. I find the paper's approach to analyzing the loss landscape of DLNs and the dynamics of SGD, along with its classification of critical points by their rank, particularly appealing. Additionally, the paper's ability to address these aspects without relying on the continuous behavior of dynamics as the learning rate approaches zero is a strong point.

**Justification For Why Not Higher Score:**

The paper could benefit from a more clear presentation of the main results.

**Justification For Why Not Lower Score:**

The paper is very strong, and there is no substantial basis for awarding a lower score.

---

### Decision · Program_Chairs · 2024-01-16

Accept (spotlight)